# REFERENCE GROUNDED SKILL DISCOVERY

**Seungeun Rho, Aaron Trinh, Danfei Xu, Sehoon Ha**
School of Interactive Computing
Georgia Institute of Technology
Atlanta, GA 30332, USA
{srho31,atrinh31,danfei,sehoonha}@gatech.edu

## ABSTRACT

Scaling unsupervised skill discovery algorithms to high-DoF agents remains challenging. As dimensionality increases, the exploration space grows exponentially, while the manifold of meaningful skills remains limited. Therefore, *semantic meaningfulness* becomes essential to effectively guide exploration in high-dimensional spaces. In this work, we present **Reference-Grounded Skill Discovery (RGSD)**[1], a novel algorithm that grounds skill discovery in a semantically meaningful latent space using reference data. RGSD first performs contrastive pretraining to embed motions on a unit hypersphere, clustering each reference trajectory into a distinct direction. This grounding enables skill discovery to simultaneously involve both imitation of reference behaviors and the discovery of semantically related diverse behaviors. On a simulated SMPL humanoid with 359-D observations and 69-D actions, RGSD successfully imitates skills such as walking, running, punching, and sidestepping, while also discover variations of these behaviors. In downstream locomotion tasks, RGSD leverages the discovered skills to faithfully satisfy user-specified style commands and outperforms imitation-learning baselines, which often fail to maintain the commanded style.

## 1 INTRODUCTION

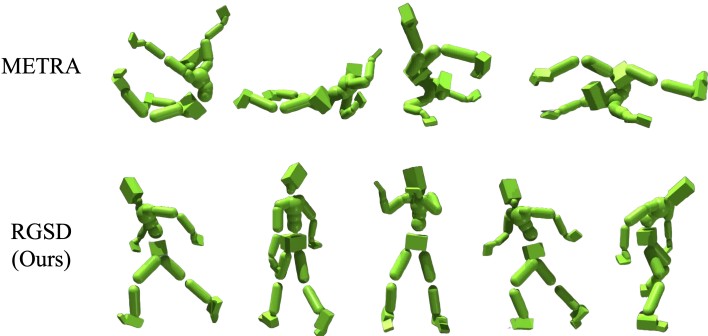

Figure 1: Comparison of skills learned by METRA and RGSD. To our knowledge, this is the first work to discover structured skills in high-DoF systems by grounding them in reference motions.

The ultimate goal of unsupervised skill discovery is to acquire a set of reusable skills that can be applied to arbitrary downstream tasks. Achieving this goal requires the learned skills to satisfy two key desiderata: *diversity* and *semantic meaningfulness*. First, the skill set should be sufficiently diverse to cover the wide distribution of possible downstream tasks. For example, a humanoid agent should be equipped with both `turn-left` and `turn-right` skills; if it only learns to `walk-forward`, it will fail on general locomotion tasks, such as goal reaching or heading. Second, the discovered skills must be semantically meaningful, as downstream tasks are typically defined in semantic terms. For example, consider a set of unstructured humanoid behaviors such as independent vibrating or

---

[1]Videos at seungeunrho.github.io/projects/RGSD

jittering of individual body parts. Although these behaviors may be diverse, it is difficult to envision realistic tasks for which such skills would be useful. Therefore, it is essential that discovered skills are not only diverse, but also structured and semantically interpretable.

Recent progress in skill discovery has demonstrated the ability to learn skills that satisfy both desiderata, particularly in relatively low degree-of-freedom (DoF) systems such as benchmark environments (Eysenbach et al., 2018; Gregor et al., 2016; Laskin et al., 2022), quadrupeds (Cheng et al., 2024; Atanassov et al., 2024; Rho et al., 2025; Cathomen et al., 2025), or simple robotic manipulators with grippers (Park et al., 2023a; Rho et al., 2024). However, scaling unsupervised skill discovery to high-DoF agents remains an open challenge. As the DoF increases, the exploration space grows exponentially, while the portion of the semantically meaningful manifold remains relatively small. This mismatch renders skill discovery in high-DoF systems particularly difficult. As shown in Fig. 1, skills learned by the state-of-the-art unsupervised skill discovery algorithm, METRA (Park et al., 2023b), fail to yield meaningful behaviors. Joints move randomly, producing highly unstructured motions in which arms, legs, torso, and head move independently and arbitrarily. This highlights the necessity of a mechanism that grounds skill discovery within a meaningful manifold.

Our key insight is that to tame the curse of dimensionality in high-DoF skill discovery, we need to construct a semantically meaningful skill latent space *a priori* and constrain exploration within this space. In this paper, we propose **Reference-Grounded Skill Discovery (RGSD)**, a simple yet effective algorithm that achieves this by leveraging reference data to ground the latent skill space before exploration begins. Building upon DIAYN (Eysenbach et al., 2018), RGSD first uses contrastive learning on reference trajectories to embed motions on a unit hypersphere, where each reference behavior corresponds to a distinct direction. This grounding step establishes a semantically meaningful manifold that constrains subsequent exploration. RGSD then utilizes this pre-structured latent space to simultaneously imitate reference skills and discover novel behaviors that are semantically related to the references. This two-stage approach is analogous to recent large language model training regimes (Guo et al., 2025; Yang et al., 2025), where pretraining with self-supervised objectives establishes a meaningful exploration space before reinforcement learning fine-tuning. As a result, RGSD not only reproduces motions present in the reference data but also discovers new skills that emerge as coherent variations of existing ones, rather than arbitrary high-dimensional movements.

We empirically demonstrate that RGSD discovers structured skills in a humanoid SMPL agent with a 359-dimensional observation space and a 69-dimensional action space. RGSD successfully imitates motions such as walking, running, side-stepping, punching, and moving backward, while also discovering related novel skills. Furthermore, it achieves superior performance on downstream tasks compared to both pure skill-discovery and imitation-based skill acquisition baselines.

The main contributions of this paper are as follows: **1)** We propose a novel skill discovery algorithm that scales to high-DoF agents by grounding the latent space with reference data. **2)** We empirically demonstrate that RGSD discovers both diverse and structured motions on a 69-DoF SMPL humanoid and achieves superior downstream task performance. **3)** We provide a theoretical proof establishing the validity of the proposed reward as a legitimate imitation signal. **4)** We offer insights into why mutual information–based methods integrate well with our approach, whereas Wasserstein dependency–based methods fail to extend in the same way.

## 2 RELATED WORKS

Our work can be framed as "imitation for discovery," because it performs imitation based on reference data to discover semantically similar yet novel behaviors. Accordingly, we first review prior work on pure **unsupervised skill discovery**, and then discuss **imitation-based skill acquisition approaches** that also learn skills by relying on reference data.

### 2.1 UNSUPERVISED SKILL DISCOVERY

The most widely adopted approach for acquiring diverse skills is based on maximizing the mutual information (MI) $\mathcal{I}(S; Z)$ between a latent variable $z$ and the agent's states $s$ (Gregor et al., 2016; Eysenbach et al., 2018; Hansen et al., 2019; Liu & Abbeel, 2021a; Laskin et al., 2022). However, because MI is typically estimated via KL divergence, it suffers from a key limitation: even minor

differences between skills can fully maximize the objective, as long as the discriminator can distinguish them. To address this, methods based on the Wasserstein Dependency Measure (WDM, Ozair et al. (2019)) have been proposed (He et al., 2022; Park et al., 2021; 2023b), which explicitly seek *maximal differences* between skills. Nevertheless, our method builds on MI-based approaches, and in Section 6 we explain why extending WDM-based algorithms with our idea faces inherent challenges. Another relevant line of work is CIC (Laskin et al., 2022), which, like our approach, employs contrastive learning. The key difference is that CIC optimizes a contrastive objective on online data gathered by the agent, leaving it with the same scaling difficulties in high-DoF systems.

More recently, approaches such as LGSD (Rho et al., 2024) and DoDont (Kim et al., 2024) have explored injecting guidance from LLMs or videos to semantically ground skill discovery. While motivated by a similar goal, these methods still struggle to scale to high-DoF systems, as their guidance is provided only at a high level. In contrast, our method leverages reference data directly, enabling a more effective grounding mechanism that scales to complex, high-DoF agents.

## 2.2 IMITATION-BASED SKILL ACQUISITION

Imitation learning offers another major paradigm for acquiring a repertoire of skills. A common approach is based on GAIL (Ho & Ermon, 2016), which uses adversarial training to align the marginal state distribution of the learned policy with that of expert demonstrations. Building on this idea, ASE (Peng et al., 2022) learns a controllable skill embedding from large-scale human motion data using a GAIL-style objective. CALM (Tessler et al., 2023) mitigates the mode collapse observed in ASE by incorporating a conditional discriminator and a motion encoder, enabling skill generation conditioned on specific reference motions. More recently, Meta-Motivo (Tirinzoni et al., 2025) combines GAIL with forward–backward representation learning (Blier et al., 2021; Touati & Ollivier, 2021) to extract a skill set tailored for zero-shot RL.

In contrast to these imitation-based approaches, we highlight a key conceptual difference: imitation objectives aim to **match the state visitation measure** of the demonstrations, whereas RGSD is a discovery algorithm that explicitly encourages visiting states *not present* from the dataset. This fundamental difference allows RGSD to discover a wider range of skills, as demonstrated in our experiments.

## 3 PRELIMINARIES

Unsupervised skill discovery can be formulated as a reward-free Markov Decision Process (MDP) $M := \{\mathcal{S}, \mathcal{A}, \mathcal{P}, \rho_0, \gamma\}$, which consists of the state space $\mathcal{S}$, the action space $\mathcal{A}$, the transition dynamics $\mathcal{P} = \Pr(s' \mid s, a)$, the initial state distribution $\rho_0$, and the discount factor $\gamma$. The objective is to *associate* a latent skill vector $z$ with the states $s$ visited by the skill-conditioned policy $\pi_\theta(\cdot \mid s, z)$, such that different $z$ induce distinct behaviors. Typically, $z \sim p(z)$ is sampled from a fixed prior at the beginning of each episode and held constant throughout.

A common formulation is to maximize the mutual information (MI) $\mathcal{I}(S; Z)$ between skills and visited states. Prior work (Gregor et al., 2016; Eysenbach et al., 2018; Laskin et al., 2022; Liu & Abbeel, 2021b) leverages the decomposition $\mathcal{I}(s; z) = \mathcal{H}(s) - \mathcal{H}(s \mid z) = \mathcal{H}(z) - \mathcal{H}(z \mid s)$. Because $\mathcal{H}(z \mid s)$ is intractable, Eysenbach et al. (2018) introduced a neural encoder $q_\phi(z|s)$ to obtain a variational lower bound $\mathcal{G}(\theta, \phi)$ of the skill discovery objective $\mathcal{F}(\theta)$:

$$\mathcal{F}(\theta) = \mathcal{I}(s; z) + \mathcal{H}(a \mid s, z)$$
$$\geq \mathbb{E}_{z \sim p(z), \, s \sim \pi(z)} \Big[ -\log p(z) + \log q_\phi(z \mid s) + \mathcal{H}\big[\pi_\theta(\cdot \mid s, z)\big] \Big] \triangleq \mathcal{G}(\theta, \phi),$$

where $\mathcal{H}[\cdot]$ denotes Shannon entropy.

Optimization proceeds as:

$$\pi_\theta : \quad \text{maximize via RL with reward } r_z = -\log p(z) + \log q_\phi(z \mid s), \tag{1}$$
$$q_\phi : \quad \text{maximize } \log q_\phi(z \mid s), \quad s \sim \pi(\cdot \mid s, z), \tag{2}$$

along with the entropy bonus for the policy. Intuitively, $q_\phi$ learns to infer which skill $z$ led to a given state, while $\pi_\theta$ is reinforced to visit states that make $z$ easily identifiable.

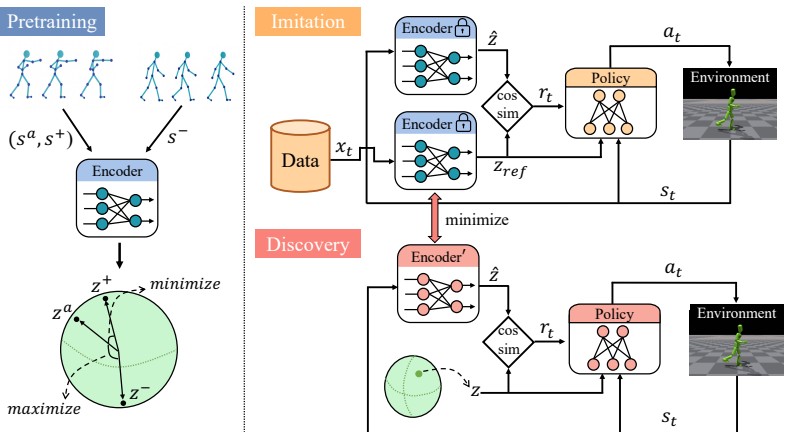

Figure 2: We present the overall training pipeline of RGSD. It starts with contrastive pretraining of an encoder using reference motions, followed by parallel training of imitation and discovery.

# 4 REFERENCE GROUNDED SKILL DISCOVERY

At a high level, RGSD approaches skill discovery in the *reverse order* of conventional methods. Standard skill discovery begins with policy exploration, and only afterward induces a latent space to enforce different regions represent different behaviors. In contrast, we start with a dataset of target behaviors we wish to capture and first embed them into a unit hyperspherical latent space without any environmental interaction. The ideal latent space would have each motion represented by a single directional vector $z$, with clear separation between different motions. This will turn exploration of the latent space into the discovery of structured skills in the state space. To achieve this, we employ contrastive learning.

Once the latent space is grounded with motion representations, we proceed with skill discovery in this reference-equipped space. Here, we exploit the geometry of the hypersphere: sampling $z$ along directions aligned with reference vectors triggers **imitation**, while sampling $z$ in-between reference directions drives the **discovery** of novel skills. This design enables semantically meaningful expansion of known behaviors and structured exploration of new ones.

In the following subsections, we detail how motion priors are injected into the latent space and how imitation and discovery are jointly achieved within this framework.

## 4.1 PRETRAINING: GROUNDING LATENT SPACE ON REFERENCE MOTIONS

The goal of pretraining is to associate each motion with a latent vector $z$, enabling the skill-conditioned policy $\pi(\cdot|s, z)$ to reproduce the corresponding trajectory. To achieve this, we place each motion at a point on the hypersphere while ensuring separation between different motions. We employ contrastive learning: positive pairs are sampled from the same trajectory, and negative pairs from different trajectories. This encourages motions to cluster tightly around unique latent directions, forming a well-structured space for downstream imitation and discovery.

More concretely, we train an encoder $q_\phi(z \mid s)$ that maps a state $s$ to a probability distribution over a latent vector $z \in \mathcal{Z}$, where $\mathcal{Z} = \{v \in \mathbb{R}^k : \|v\|_2 = 1\}$. We model $q_\phi(z \mid s)$ with a von Mises–Fisher (vMF) distribution:

$$q_\phi(z \mid s) \; \propto \; \exp\big(\kappa\, \mu_\phi(s)^\top z\big), \tag{3}$$

where $\mu_\phi(s)$ is the mean direction parameterized by a neural network $\phi$, and $\kappa$ is the concentration parameter. We fix $\kappa$ as a constant during training.

Training is performed with contrastive learning on a reference dataset $\mathcal{M} = \{m_i\}_{i=1}^n$, where each $m_i$ is a reference trajectory consisting of a sequence of states. During pretraining, we first sample a motion $m \sim \mathrm{Uniform}(\mathcal{M})$, then select a pair of states $(s^a, s^+)$ from $m$, with $s^a$ as the anchor and $s^+$ as the positive sample. A negative sample $s^-$ is drawn from $\mathcal{M} \setminus \{m\}$. Given the anchor $s^a$, the

positive $s^+$, and negatives $\{s_j^-\}$, we embed them as

$$z^a = \mu_\phi(s^a), \quad z^+ = \mu_\phi(s^+), \quad z_j^- = \mu_\phi(s_j^-),$$

and optimize the InfoNCE loss (Oord et al., 2018):

$$\mathcal{L}_{\text{InfoNCE}} = -\log \frac{\exp\big(\text{sim}(z^a, z^+)/T\big)}{\exp\big(\text{sim}(z^a, z^+)/T\big) + \sum_j \exp\big(\text{sim}(z^a, z_j^-)/T\big)}. \tag{4}$$

Here, $\text{sim}(z_i, z_j) = z_i^\top z_j$ denotes cosine similarity since all vectors lie on the unit hypersphere, and $T = 1/\kappa$ is the temperature. A detailed derivation is provided in Appendix B.

As depicted in Fig.2-pretraining, the mapping of motions in latent space is spread out before training. However when pretraining is completed, we end up with a within-motion alignment, meaning that the mapping of every state $s \in m$ in each motion $m$ points in exactly the same direction. This characteristic is important for the following section; therefore, for completeness, we provide a proof of this claim in Appendix C.

## 4.2 IMITATION OF REFERENCE SKILLS

After pretraining, we freeze $q_\phi$ and proceed to the second phase, where imitation and discovery are trained in parallel. Interestingly, both processes rely on the **same reward term** defined in Eq. 1. We first show that this reward can be repurposed as an *imitation reward* when conditioned on the motion's embedding of the pretrained encoder $q_\phi$. Specifically, we sample a motion $m \sim \text{Uniform}(\mathcal{M})$ and compute its embedding vector $z_m$ as the average of its state embeddings:

$$z_m = \frac{1}{l} \sum_{s \in m} \mu_\phi(s), \tag{5}$$

where $l$ is the length of motion $m$. At theoretical optimum, $z_m$ should be aligned with the embedding of any individual state $s \in m$.

Conditioning the policy on $z_m$, i.e., executing rollouts with $\pi(\cdot|s, z_m)$, does not by itself reproduce $m$. However, reinforcement learning with the DIAYN reward (Eq. 1) drives $\pi(\cdot|s, z_m)$ toward imitating $m$. Substituting in the vMF formulation (Eq. 3) gives

$$r(s, z_m) = -\log p(z) + \log q_\phi(z_m \mid s) \tag{6}$$
$$= C + \kappa\, \mu_\phi(s)^\top z_m, \tag{7}$$

where the prior $p(z)$ and concentration $\kappa$ are fixed, and $\phi$ is frozen. $C$ is a constant term independent of $s$ and $z_m$. Intuitively, this reward depends on the angle between $\mu_\phi(s)$ and $z_m$, effectively measuring the similarity between the state visited by the agent and the states in the reference motion. Moreover, initializing $s_0$ from $m$ provides an aligned starting point; as trajectories deviate, the cosine alignment decreases and the reward diminishes.

This formulation can be viewed as *feature-based imitation* with structured representations, as proposed by Li et al. (2025). Unlike DeepMimic (Peng et al., 2018) or MaskedMimic (Tessler et al., 2024), which reward joint-level similarity, our method evaluates similarity in a learned latent space.

**Guarantee as an imitation reward**  For the proposed reward to serve as a valid imitation objective, it must satisfy two key conditions: (1) it should attain its optimum when the agent visits the exact states of the reference motion, and (2) the reward landscape around the reference states should be locally quasi-concave, ensuring that deviations from the reference lead to a monotonic decrease in the reward within a neighborhood of the optimum. We claim that, under the assumption of "perfect alignment within motion" introduced in the previous section, our reward formulation in Eq. 7 satisfies both conditions, and we provide the proof in Appendix D.1. To exploit this local concavity in practice, we apply *early termination*: whenever the agent deviates from the reference motion beyond a specified threshold measured by cartesian error, the episode is terminated. This mechanism guarantees that the reward functions as a legitimate imitation objective.

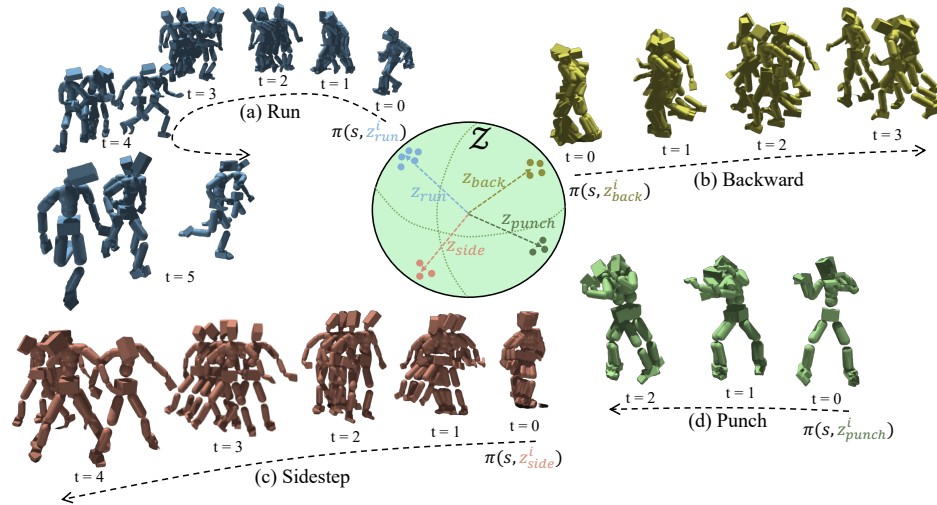

Figure 3: **Example skills.** RGSD generates diverse behaviors when conditioned on different latent vectors. The figure shows motions from a **single policy** conditioned on distinct latent vectors sampled near the embedding of the **(a)** running, **(b)** backward, **(c)** sidestepping, and **(d)** punching motion.

### 4.3 DISCOVERY OF NEW SKILLS

The discovery process follows DIAYN, but with several key differences. First, to protect the learned latent space during discovery, we initialize a separate encoder $q'_\phi$ from the frozen $q_\phi$ and continue propagating gradients from minimizing KL-divergence between $q'_\phi$ and $q_\phi$. Second, we train discovery in parallel with imitation, so that the shared policy and value function transfers knowledge about high fidelity behavior from imitations into discovery. Note that these two processes share same form of reward function and latent space, so these shared components can be optimized in a stable manner. Third, we adopt reference state initialization (RSI), which samples initial states directly from the reference motions. RSI prevents the emergent of *disjoint* skill sets by ensuring that imitation and discovery operate over overlapping state distributions.

In detail, imitation and discovery are trained in parallel, with a ratio parameter $p$:

$$z \sim \begin{cases} \mu_\phi^-(m), & \text{with probability } p, \\ k/\|k\|, & k \sim \mathcal{N}(0, I), & \text{with probability } 1-p, \end{cases}$$

where $m$ is the motion sampled by RSI. The full algorithmic details are provided in Appendix F.

## 5 EXPERIMENTAL RESULTS AND DISCUSSION

In this section, we investigate four questions: (1) Can RGSD imitate reference motions with high fidelity? (2) Can RGSD discover novel skills that are still semantically related to the references? (3) Can we control degree of diversity at test time? (4) Can the learned skills be effectively leveraged for downstream tasks?

**Baselines.** We compare RGSD against both unsupervised skill discovery (USD) methods and imitation learning (IL) based skill acquisition methods. To assess the impact of reference guidance, we include pure USD baselines. Specifically, we compare against DIAYN (Eysenbach et al., 2018) and METRA (Park et al., 2023b). DIAYN, which forms the backbone of RGSD, also serves as an ablation without reference grounding, while METRA represents a distance-maximization approach to USD that explicitly seeks maximal differences between skills.

For the IL baselines, we consider ASE, CALM, and Meta-Motivo, all of which utilize a GAIL-style objective (Ho & Ermon, 2016) for imitation. ASE builds on InfoGAIL (Li et al., 2017), combining mutual-information maximization with an adversarial imitation reward, making it the closest baseline that jointly addresses skill discovery and imitation. CALM, a variant of ASE, further augments

Table 1: Cartesian ERR(cm) and FID scores for imitation across tasks. Lower values are better.

| | Walk | | Run | | Sidestep | | Backward | | Punch | |
|---|---|---|---|---|---|---|---|---|---|---|
| | ERR | FID | ERR | FID | ERR | FID | ERR | FID | ERR | FID |
| DIAYN | 46.7 | 70.1 | 52.8 | 132.0 | 27.4 | 23.7 | 36.7 | 72.8 | 50.7 | 53.2 |
| METRA | 42.0 | 67.6 | 51.8 | 140.3 | 44.7 | 32.8 | 47.4 | 75.7 | 51.5 | 40.9 |
| ASE | 8.2 | 2.9 | 16.4 | 11.3 | 10.3 | 22.2 | 11.6 | 62.6 | 9.0 | 4.9 |
| CALM | **7.2** | **1.4** | 15.0 | 13.9 | 11.8 | 3.2 | 10.1 | 45.8 | 9.2 | 4.2 |
| Meta-Motivo | 10.9 | 1.7 | 15.4 | **4.9** | 11.8 | **1.4** | 8.6 | **2.1** | 8.1 | **2.8** |
| **RGSD (Ours)** | 7.4 | 4.7 | **7.7** | 9.4 | **6.7** | 8.0 | **6.7** | 4.0 | **7.7** | 8.8 |

the imitation objective using a motion-conditioned discriminator. Meta-Motivo integrates imitation rewards with forward–backward representation learning.

We emphasize that while these methods rely on GAIL-based imitation rewards, our approach employs a novel imitation reward derived from the DIAYN objective. To ensure a clean algorithmic comparison, we remove additional engineering techniques used in the original ASE and CALM implementations. The exact modifications are described in Appendix G.2.

**Experimental setup.** We selected 20 reference motions from the ACCAD dataset. These motions are categorized into five tasks: `walk`, `run`, `sidestep`, `backward`, and `punch`, with each task containing 2 to 6 relevant motions. A complete list of motions is provided in Appendix G.3. Training was conducted in the GPU-based simulator Isaac Gym (Makoviychuk et al., 2021), using PPO (Schulman et al., 2017) as the RL algorithm. The full set of hyperparameters is provided in Appendix G.4. To ensure stable training, all methods employed an early termination condition: whenever the robot fell, the episode was terminated.

## 5.1 EVALUATION OF IMITATION

We first assess how well RGSD reproduces reference motions. For each motion, the agent is initialized with the first state $s_0$ of the motion. Then we condition the policy on $(s_0, z_m)$, where $z_m$ is computed with Eq. 5. We generate 500 trajectories per each motion and compared against the reference to compute two metrics:

- **Cartesian error:** the average $\ell_2$ distance between corresponding body-part positions per frame, averaged over the trajectory.
- **Motion FID:** the Fréchet distance between Gaussian feature distributions fitted to reference motions and generated motions. Lower FID indicates better naturalness.

For CALM and Meta-Motivo, selecting the right latent that represents each motion is straightforward since it also includes motion encoders. For methods without encoders, we uniformly sample 500 latent vectors, select the one that minimizes Cartesian error, and re-evaluate using this vector to ensure fairness. We found that 500 samples were sufficient, as increasing the number further did not yield noticeable improvements.

**RGSD achieves high-fidelity imitation.** As shown in Table 1, RGSD achieves low Cartesian error while Meta-Motivo achieves low FID scores across most motions. In contrast, pure USD baselines fail to discover skills that closely resemble the reference dataset. While DIAYN has demonstrated the ability to discover basic locomotion behaviors in low DoF ranging from 3 to 6 environments such as HalfCheetah, Ant, and Hopper (Brockman et al., 2016), they struggle with the 69-DoF SMPL agent, producing behaviors that are not semantically meaningful. This emphasizes the impact of reference guidance.

When compared to Meta-Motivo, the results highlight a trade-off between fidelity and naturalness. Meta-Motivo achieves lower FID scores in 4 out of 5 tasks, indicating that its motions generally appear more natural. However, RGSD outperforms Meta-Motivo in Cartesian error on 4 out of 5 tasks, demonstrating higher trajectory fidelity. This contrast reflects a key difference in objective design: Meta-Motivo employs an occupancy-matching objective at the motion level, leading to smoother but less precise reproductions, whereas RGSD relies on frame-level similarity rewards, yielding high fidelity motions.

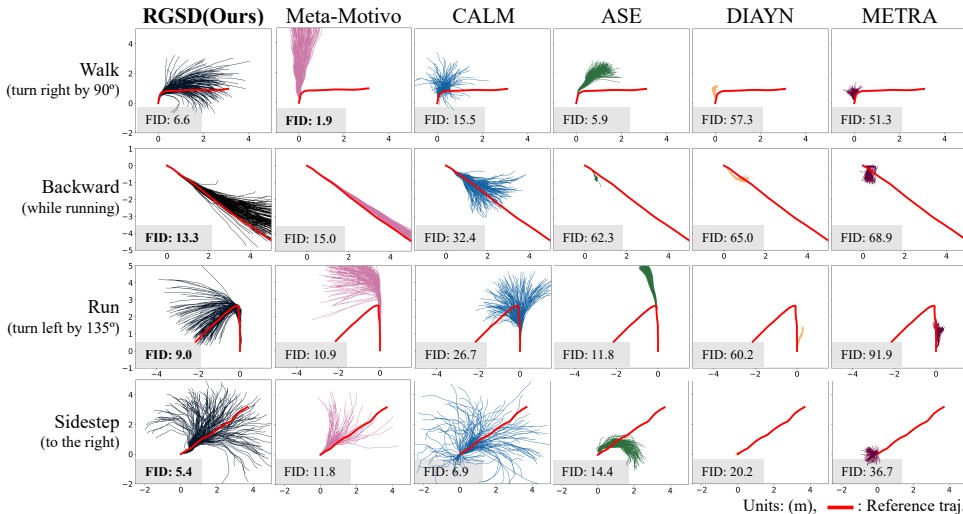

Figure 4: Top-view trajectories of the robot's base when conditioned on latent vectors sampled from the neighborhood of each motion embedding. For each method, we visualize 150 trajectories.

## 5.2 EVALUATION OF DISCOVERED SKILLS

Now we evaluate whether RGSD can discover skills that are semantically similar to existing ones while also producing novel variations. Semantic similarity is enforced by sampling latent vectors from a local neighborhood around each motion embedding using a vMF distribution with the fixed concentration parameter $\kappa$ at 50.

**RGSD can discover semantically similar yet novel behaviors.** Fig. 3 illustrates behaviors generated by the policy when conditioned on sampled latent vectors. The method produces diverse behaviors that remain semantically consistent with the reference motions. For example, while the reference dataset contains only a **single sidestepping** motion to the right, RGSD discovers skills that preserve the sidestepping style but introduce diverse degree of turns, enabling sidestepping in multiple directions. Similarly, although the dataset includes a punching motion aimed at a fixed target, our method extends this behavior to generate punches toward a variety of target positions.

To compare against baselines, we visualize the 150 top-view trajectories of the robot's base for each method in Fig. 4, alongside the trajectories from the reference motions. Trajectories produced by RGSD remain tightly clustered around the references, indicating that the policy captures the intended motion structure. In contrast, the baseline methods often produce degenerate or drifting trajectories, revealing their inability to preserve the characteristics of the original movements.

These observations are further supported by the FID scores. For three out of four motions, RGSD achieves the lowest FID. While Meta-Motivo typically achieves the second-lowest FID, its generated trajectories still fail to remain centered around the references. CALM shows particularly degraded performance: its FID increases from 1.4 to 15.5 for walking and from 13.9 to 26.7 for running, indicating difficulty in producing diverse yet high-fidelity behaviors. In contrast, the FID scores of RGSD remain stable, demonstrating that the method successfully preserves motion style while enabling variation.

We attribute this difference to the training setup. In both Meta-Motivo and CALM, the policy is always conditioned directly on the motion embedding and therefore never encounters nearby latent vectors during training. By contrast, RGSD explicitly separates imitation and discovery phases: the imitation phase exposes the policy to motion-embedding latents, while the discovery phase conditions it on diverse samples from the latent neighborhood. Consequently, the policy learns to handle both exact motion embeddings and their local variations, enabling consistent and semantically meaningful skill discovery.

**RGSD enables test-time control over behavioral diversity.** A key advantage of RGSD is that it allows users to modulate the diversity of generated behaviors *at test time* by adjusting the sampling distribution of the latent vector $z$. Recall that we sample $z \sim \text{vMF}(z_m, \kappa)$, a von Mises–Fisher distribution centered at the motion embedding $z_m$ with concentration parameter $\kappa$. A larger $\kappa$ produces

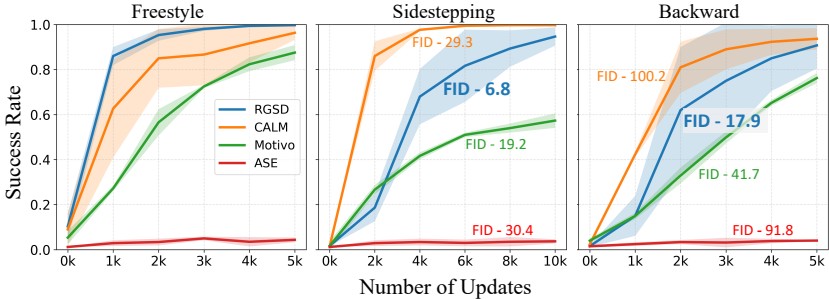

Figure 5: Training curves for the downstream task, along with corresponding FID scores.

samples tightly clustered around $z_m$, yielding behaviors that closely imitate the reference motion but exhibit lower diversity. Conversely, a smaller $\kappa$ yields a broader distribution, producing more diverse behaviors that may deviate further from the original motion $m$.

We evaluate this effect on three reference motions by sampling with $\kappa \in \{20, 100, 1000\}$. Results are shown in Fig. 6. As illustrated, increasing $\kappa$ results in trajectories that stay stylistically similar to the reference motion with limited variation, while decreasing $\kappa$ produces increasingly diverse trajectories that still preserve the core semantic style.

### 5.3 DOWNSTREAM TASK EVALUATION

**Setup.** We consider the `GoalReaching` task with three style conditions: `freestyle`, `sidestepping`, and `backward`. The agent must reach a randomly spawned goal within a $20 \times 20$ m arena while adhering to the specified style. Following the setup in the previous sections, we first train a low-level policy $\pi_{\text{low}}(\cdot \mid s, z)$ using the 20 reference motions for each method. We then train a high-level policy $\pi_{\text{high}}(z \mid s, g)$, which takes the goal $g$ as input and outputs a high-level latent action $z$. The corresponding low-level policy $\pi_{\text{low}}(\cdot \mid s, z)$ then produces the joint action $a$. During this phase, $\pi_{\text{low}}$ is frozen, and we optimize only $\pi_{\text{high}}$.

To encourage both goal reaching and adherence to the commanded style, we adopt a reward similar to Tessler et al. (2023):

$$r = \exp(-0.25 * \|d_{\text{goal}}\|) + \exp(-8 * \|z \oplus z_m\|),$$

where $d_{\text{goal}}$ is the distance to the goal, and $z \oplus z_m$ denotes the cosine distance between the high-level action $z$ and the embedding $z_m$ of the directed motion. For the `freestyle` task, only the first reward term is applied. All experiments are conducted with three random seeds.

**Only RGSD consistently respects the commanded style while reaching the goal.** Figure 5 reports the success rates and FID scores of RGSD and baseline methods over the course of training. In the `freestyle` setting, since no specific style is required, we evaluate only the success rate. Although all methods eventually achieve near-perfect success, RGSD converges significantly faster. In contrast, ASE frequently fails to reach the goal; we observed that it often remains stationary, exploiting the fact that its reward remains positive even without producing the desired motion.

For the `sidestepping` and `backward` tasks, we evaluate both the success rate and the FID score. Only RGSD reliably follows the commanded style. Although CALM achieves higher success rates early in training, RGSD eventually matches it. Furtermore, CALM ultimately ignores the style command and simply runs forward. On the other hand, Meta-Motivo respects the style only under certain conditions. For example, in the backward task, it walks backward when the goal is spawned behind the agent, but when the goal appears in front, the agent decides to walk forward. In contrast, RGSD consistently adheres to the commanded style: even when the goal is spawned in front during backward task, it takes a large detour to maintain backward motion. This difference is reflected in the FID scores: Meta-Motivo, CALM, and ASE all exhibit higher FID values, indicating substantial deviations from the intended styles. We refer the reader to the accompanying video for qualitative examples.

Note that these tasks are inherently challenging because the reference motions for sidestepping and backward running do not include any turning behaviors. To reach diverse goal locations while main-

taining style fidelity, an agent must discover additional skills that are semantically consistent with the style, such as backward turns at varying angles. Imitation-based baselines lack this flexibility, whereas RGSD acquires a rich set of turning behaviors, enabling it to reach goals while preserving the commanded style.

# 6 UNDERSTANDING THE CHALLENGES OF EXPANDING RGSD WITH METRA

Our backbone skill discovery module is DIAYN, which relies on maximizing mutual information (MI). However, a well-known limitation of MI-based objectives is that they can be fully optimized even when the underlying differences between skills are minimal, as long as the discriminator can distinguish them. This raises a natural question: can we extend RGSD on top of distance-maximization based approaches such as METRA?

To explore this, we conduct experiments with a METRA-based variant of RGSD. Similar to DIAYN, we first ground the latent space with reference motions while respecting the two conditions that METRA enforces. First, each motion should align with a unique directional vector $z$, which can be satisfied using the original contrastive loss. Second, the latent space should capture the notion of *temporal distance*: $\forall x, y \in S_{adj}, \|\phi(x) - \phi(y)\| \leq 1$. To enforce this, we add a loss term that drives $\|\phi(x) - \phi(y)\|$ toward its maximum value of 1 for adjacent state pairs. The resulting latent space, shown in Fig. 9, places each motion along a distinct line.

However, this formulation exposes a critical issue when dealing with **repetitive motions**. Consider walking: the agent begins at some pose, takes a step, and eventually returns to a pose nearly identical to the starting one. Because all observations are computed in the local frame, the initial state $s_0$ and final state $s_T$ become identical. In this case, the METRA reward $(\phi(s_T) - \phi(s_0))^\top z$ collapses to zero. As a result, repetitive behaviors cannot be framed as reward maximization under METRA, and more broadly, WDM-based approaches defined in a local coordinate system are not straightforward to capture repetitive dynamics. This observation is consistent with the concurrent findings (Park et al.).

One might suspect that this problem could be bypassed by augmenting the state with additional variables, such as a time variable or global coordinates. However, this does not resolve the issue. In practice, it causes the latent space to be dominated by these added dimensions, leading the policy to focus on them rather than discovering meaningful behaviors. For brevity, we defer a more detailed discussion to Appendix H. To properly represent a substantial portion repetitive motions in dataset, we choose DIAYN as our backbone.

# 7 CONCLUSION

We introduced **Reference-Grounded Skill Discovery** (RGSD), a simple yet effective framework that scales unsupervised skill discovery to high-DoF agents by *grounding* exploration in a semantically meaningful latent space constructed from reference motions. On a 69-DoF SMPL humanoid with 359-D observations, RGSD reproduces complex motions, such as walking, running, sidestepping, backward walking, punching, with high fidelity, and also discovers coherent variants, outperforming state-of-the-art unsupervised discovery and imitation-based baselines on both motion metrics and downstream control.

Despite these advancements, several avenues remain open for further exploration. One promising direction is to move beyond variants of individual skills toward genuinely *compositional* behaviors and principled interpolations that blend primitives (e.g., "walking while punching"). Another interesting direction would be scaling across embodiments and datasets, with the long-term vision of building a skill foundation model for control, analogous to large language models in natural language processing. Overall, we believe our work represents the beginning of a practical recipe for scaling skill discovery in high-DoF agents through reference grounding.

## REPRODUCIBILITY STATEMENT

We have made extensive efforts to ensure the reproducibility of our work across multiple dimensions:

- A complete pseudo-code description of our algorithm is provided in Appendix F.
- The full list of reference motions used in our experiments is detailed in Appendix G.3.
- Comprehensive specifications of states, actions, and hyperparameters are given in Appendix G.4.
- Rigorous derivations and proofs of our theoretical claims are included in Appendices B, C, and D.1.

## ACKNOWLEDGMENTS

This research has been funded by the Industrial Technology Innovation Program (P0028404, development of a product level humanoid mobile robot for medical assistance equipped with bidirectional customizable human-robot interaction, autonomous semantic navigation, and dual-arm complex manipulation capabilities using large-scale artificial intelligence models) of the Ministry of Industry, Trade and Energy of Korea. In addition, we thank Jeonghwan Kim for the discussion.

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

## A QUALITATIVE RESULTS FOR CONTROLLING DIVERSITY EXPERIMENT

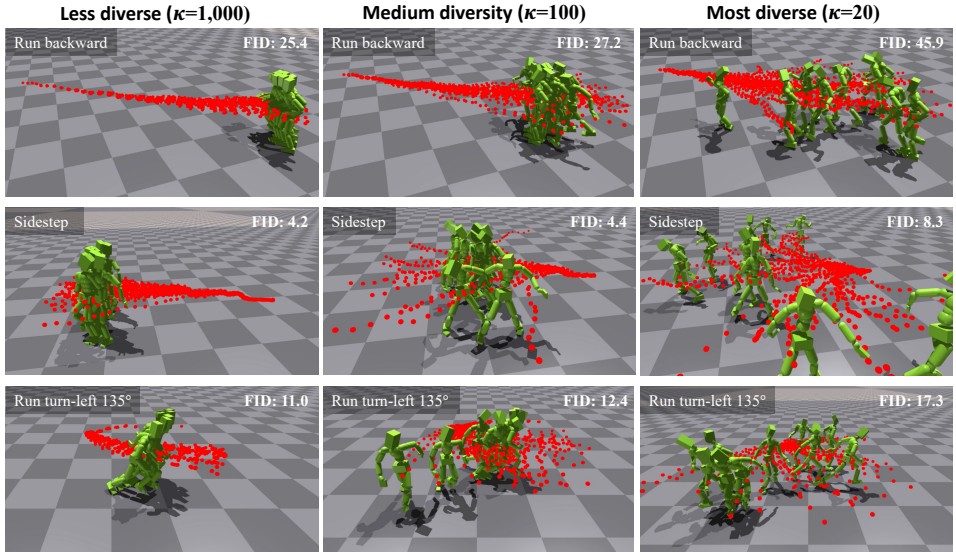

Figure 6: Red dots indicate the agent's trajectory. From left to right, as we sample the latent variable $z$ from increasingly wider distributions centered around the reference motion, the resulting behaviors become more diverse while still preserving the core characteristics of the original motion.

## B FROM vMF ENCODER TO INFONCE LOSS

Recall the encoder defines a von Mises–Fisher (vMF) density on the unit hypersphere:

$$q_\phi(z \mid s) = C_d(\kappa) \, \exp\!\big(\kappa \, \mu_\phi(s)^\top z\big), \quad \|z\|_2 = \|\mu_\phi(s)\|_2 = 1,$$

where $C_d(\kappa)$ is the normalizing constant and $\kappa > 0$ is fixed.

Given an anchor $s^a$ and a candidate set $\mathcal{C} = \{z^+, z_1^-, \ldots, z_K^-\}$ containing one positive and $K$ negatives, consider the conditional probability that the positive $z^+$ was generated from $q_\phi(\cdot \mid s^a)$ among the candidates:

$$p_\phi\big(\text{pos} = z^+ \mid s^a, \mathcal{C}\big) = \frac{q_\phi(z^+ \mid s^a)}{\sum_{z \in \mathcal{C}} q_\phi(z \mid s^a)} = \frac{C_d(\kappa) \exp\!\big(\kappa \, \mu_\phi(s^a)^\top z^+\big)}{\sum_{z \in \mathcal{C}} C_d(\kappa) \exp\!\big(\kappa \, \mu_\phi(s^a)^\top z\big)}.$$

Because $\kappa$ is fixed, $C_d(\kappa)$ cancels:

$$p_\phi\big(\text{pos} = z^+ \mid s^a, \mathcal{C}\big) = \frac{\exp\!\big(\kappa \, \mu_\phi(s^a)^\top z^+\big)}{\exp\!\big(\kappa \, \mu_\phi(s^a)^\top z^+\big) + \sum_{j=1}^{K} \exp\!\big(\kappa \, \mu_\phi(s^a)^\top z_j^-\big)}.$$

Maximizing the log-likelihood of the correct positive is therefore equivalent to minimizing the cross-entropy loss

$$\mathcal{L}_{\text{NCE}} = -\log \frac{\exp\!\big(\kappa \, \mu_\phi(s^a)^\top z^+\big)}{\exp\!\big(\kappa \, \mu_\phi(s^a)^\top z^+\big) + \sum_{j=1}^{K} \exp\!\big(\kappa \, \mu_\phi(s^a)^\top z_j^-\big)}.$$

Identifying the *similarity* as the dot product on the unit sphere, $\text{sim}(u, v) = u^\top v$ (cosine similarity), and setting the temperature $T = 1/\kappa$, we obtain the standard InfoNCE form:

$$\mathcal{L}_{\text{InfoNCE}} = -\log \frac{\exp\!\big(\text{sim}(\mu_\phi(s^a), z^+)/T\big)}{\exp\!\big(\text{sim}(\mu_\phi(s^a), z^+)/T\big) + \sum_{j=1}^{K} \exp\!\big(\text{sim}(\mu_\phi(s^a), z_j^-)/T\big)}. \tag{8}$$

If we instantiate the candidates by their mean directions, i.e., $z^a = \mu_\phi(s^a)$, $z^+ = \mu_\phi(s^+)$, and $z_j^- = \mu_\phi(s_j^-)$, then the loss reduces exactly to the temperature-scaled dot-product InfoNCE used in the main text (with $T = 1/\kappa$).

## C   PROOF OF WITHIN–MOTION ALIGNMENT

We prove that under the InfoNCE objective with "same–motion" positives, the optimal encoder perfectly aligns all frames of a given motion to a single unit vector on the sphere.

**Proof.**   Let there be $N$ motions. Each motion $m \in \{m_1, \ldots, m_N\}$ is a sequence of states $s \in \mathbb{R}^d$. An encoder $\mu_\phi$ maps each state to a unit vector $u = \mu_\phi(s) \in \mathcal{Z}$.

For an anchor $u \in m$, we draw a positive $v \in m$ (independent) and negatives $n_1, \ldots, n_K$ from other motions. The InfoNCE loss for this triplet is

$$\ell(u, v, \{n_j\}) = -\log \frac{\exp(u^\top v/T)}{\exp(u^\top v/T) + \sum_{j=1}^K \exp(u^\top n_j/T)}. \tag{9}$$

Fix $u$ and negatives $\{n_j\}$. Write $s = u^\top v \in [-1, 1]$ and $C = \sum_j \exp(u^\top n_j/T)$. Then

$$\ell(s) = -\frac{s}{T} + \log\big(\exp(s/T) + C\big).$$

Differentiating LHS with $s$,

$$\frac{d\ell}{ds} = \frac{1}{T}\Big(-\frac{C}{\exp(s/T) + C}\Big) < 0,$$

so $\ell(s)$ **is monotonically decreasing in** $s$. Hence, the loss is minimized by maximizing $u^\top v$, which is at most 1 and attained only when $v = u$. This implies that the angle between embeddings of frames from the same motion should be minimized, resulting in a perfect alignment.

**Reduction of the loss.**   After the alignment, for motion $m$ we have $u = v = \mu_m$, and equation 9 reduces to

$$\ell_m = -\log \frac{e^{1/T}}{e^{1/T} + \sum_{j \neq m} e^{\mu_m^\top \mu_j/T}}.$$

The total loss $\sum_m \ell_m$ now depends only on the pairwise inner products $\mu_m^\top \mu_j$. Minimizing this objective pushes the vectors $\{\mu_m\}$ as far apart as geometry allows, yielding a regular simplex when $M \leq k + 1$.

## D   PROOF OF GUARANTEE AS AN IMITATION REWARD

Our goal is to prove two properties of the reward function in Eq. 7: **(i)** the reward achieves its optimum when the agent visits the exact states of motion $m$, and **(ii)** the reward function is locally quasi-concave around the states $s \in m$.

### D.1   PROOF OF REWARD OPTIMALITY ON MOTION STATES

**Assumption.**   We assume that the pretraining of $\mu_\phi$ in Section 4.1 has reached its theoretical optimum. More concretely, for all $m \in \mathcal{M}$ and for all $\{s_1, s_2, \ldots, s_{l_m}\} \subset m$, we have

$$\mu_\phi(s_1) = \mu_\phi(s_2) = \cdots = \mu_\phi(s_{l_m}),$$

where $l_m$ is the number of states in motion $m$. Therefore, from Eq. 5,

$$z_m = \frac{1}{l_m} \sum_{s \in m} \mu_\phi(s) = \frac{1}{l_m} l_m \mu_\phi(s_i) = \mu_\phi(s_i), \quad \forall i. \tag{10}$$

**Proof.** From Eq. 7, the reward can be written as

$$r(s, z_m) = -\log p(z) + \log q_\phi(z_m \mid s) \tag{11}$$

$$= C + \kappa \mu_\phi(s)^\top z_m, \tag{12}$$

where $C$ and $\kappa$ are constants that do not affect the optimal point. Discarding the constants, we can write

$$r(s, z_m) = \langle \mu_\phi(s), z_m \rangle.$$

Substituting $\mu_\phi(s)$ with Eq. 10, we obtain

$$\forall s \in m, \, r(s, z_m) = \langle \mu_\phi(s), z_m \rangle = \langle z_m, z_m \rangle = 1.$$

Since $r(s, z_m)$ is bounded within $[-1, 1]$, this value is the global optimum. Hence, every state $s \in m$ achieves the global optimum.

## D.2 PROOF OF LOCAL QUASI-CONCAVITY OF THE REWARD

**Assumption.** We assume from Eq. 10 that for every $s^\star \in m$ we have $\mu_\phi(s^\star) = z_m$. We designed the neural network $\hat{\mu}_\phi$ to be piecewise linear in a neighborhood of $s^\star$, and normalization is applied to the output so that $\mu_\phi(s) = \hat{\mu}_\phi(s)/\|\hat{\mu}_\phi(s)\|$ lies in a unit hypersphere for all $s \in \mathcal{S}$.

**Lemma.** Let $s^\star \in \mathcal{S}$ be a state in a motion that satisfies $\mu_\phi(s^\star) = \hat{\mu}_\phi(s^\star)/\|\hat{\mu}_\phi(s^\star)\|_2 = z_m$. Then $r(s) := \langle \mu_\phi(s), z_m \rangle$ is quasi-concave in a neighborhood of $s^\star$.

**Proof.** To show that $r(s)$ is quasi-concave in a neighborhood of $s^\star$, we want to show that inside the neighborhood $B_\epsilon(s^\star)$, the superlevel set $S_\alpha(r) = \{s : r(s) \geq \alpha\}$ is convex.

To show this, we first check the convexity of the superlevel set in the latent space. Then, by leveraging the fact that the preimage of an affine transform preserves convexity, we conclude the convexity of the superlevel set in the state space.

**Convexity of superlevel set in latent space.** We define a function in latent space $f(z) = z_m^\top z/\|z\|_2$. This satisfies $f(\hat{\mu}_\phi(s)) = r(s)$. We now show the convexity of the superlevel set of $f$:

$$S_\alpha(f) = \left\{ z : \, f(z) = z_m^\top \frac{z}{\|z\|_2} \geq \alpha \right\}$$
$$= \left\{ z : \, z_m^\top z \geq \alpha \|z\|_2 \right\}.$$

To show its convexity, for $\forall z_1, z_2 \in S_\alpha(f)$ and $t \in [0, 1]$, we plug in $z \leftarrow tz_1 + (1 - t)z_2$:

$$z_m^\top(tz_1 + (1 - t)z_2) = tz_m^\top z_1 + (1 - t)z_m^\top z_2$$
$$\geq t\alpha\|z_1\|_2 + (1 - t)\alpha\|z_2\|_2$$
$$= \alpha\big(\|tz_1\|_2 + \|(1 - t)z_2\|_2\big)$$
$$\geq \alpha\|tz_1 + (1 - t)z_2\|_2.$$

Therefore, $tz_1 + (1 - t)z_2 \in S_\alpha(f)$, so $S_\alpha(f)$ is convex.

**Convexity of superlevel set in state space.** Here, we use the fact that the preimage of an affine transform also preserves convexity: when $T(s) := As + b$ is affine, then $T^{-1}(S_\alpha(f))$ is convex.

The superlevel set of $r(s) = f(\hat{\mu}_\phi(s))$ is defined as:

$$S_\alpha(r) = \big\{ s : r(s) \geq \alpha \big\}$$
$$= \big\{ s : f(\hat{\mu}_\phi(s)) \geq \alpha \big\}.$$

Since the affine property only applies near the neighborhood of $s^\star$, we can define an open ball around $s^\star$ that satisfies such a condition:

$$\hat{\mu}_\phi(s) = As + b \qquad \text{for all } s \in B_\epsilon(s^\star) := \{s : \|s - s^\star\|_2 < \epsilon\}.$$

Restricting the domain to the ball, we have:

$$
\begin{aligned}
\{s \in B_\epsilon(s^\star) : r(s) \geq \alpha\} &= S_\alpha(r) \cap B_\epsilon(s^\star) \\
&= \{s : r(s) \geq \alpha\} \cap B_\epsilon(s^\star) \\
&= \{s : f(\hat{\mu}_\phi(s)) \geq \alpha\} \cap B_\epsilon(s^\star) \\
&= \{s : f(T(s)) \geq \alpha\} \cap B_\epsilon(s^\star) \\
&= T^{-1}(\{z : f(z) \geq \alpha\}) \cap B_\epsilon(s^\star) \\
&= T^{-1}(S_\alpha(f)) \cap B_\epsilon(s^\star).
\end{aligned}
$$

This set is also convex because the intersection of two convex sets is also convex. This proves that the superlevel set of $r(\cdot)$ is convex local to $s^\star$.

**Conclusion**  By definition of quasi-concavity, we conclude that $r(\cdot)$ is quasi-concave in the neighborhood of $s^\star$, $B_\epsilon(s^\star)$.

# E WITHIN-MOTION ALIGNMENT: LATENT SPACE VISUALIZATION AFTER PRETRAINING

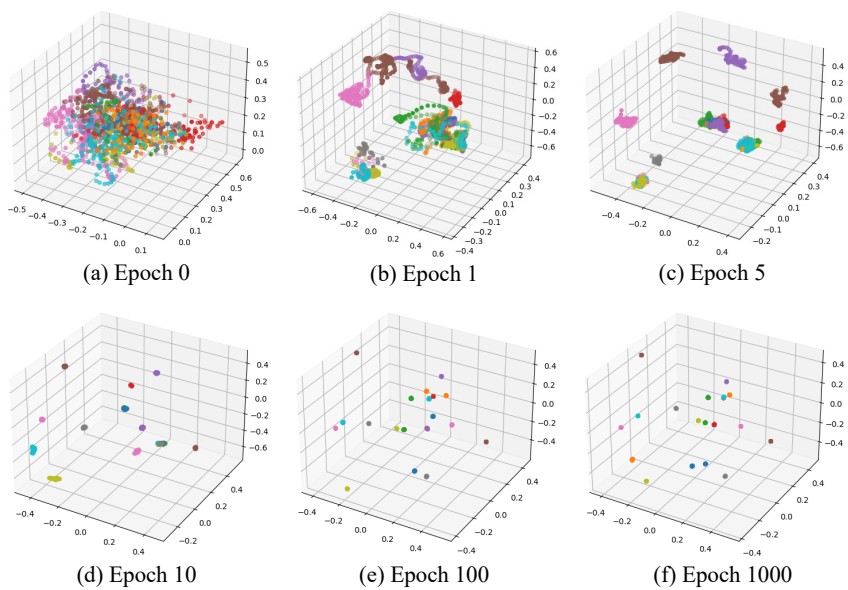

Figure 7: Latent embeddings of all 20 reference motions using the pretrained encoder. Each point corresponds to a single frame, and different colors represent different motions.

To evaluate the degree of within-motion alignment, we visualize the latent embeddings of each motion in a 3D space using the pretrained encoder $\mu_\phi$. Because the latent vectors are 16-dimensional, we display only their first three dimensions. As shown in Fig. 7, after 1000 epochs of training with the loss in Eq. 8, the state-wise embeddings within each motion form tight clusters and are nearly perfectly aligned, which is consistent with our theoretical results in Section C.

We also quantify this alignment using cosine similarity. For each motion $m_i \in \mathcal{M}$, we first compute its motion-level embedding $z_{m_i}$ using Eq. 5. Then, for every state $s \in m_i$, we measure the cosine similarity between its latent embedding and $z_{m_i}$, and average these values to obtain the within-motion alignment score for that motion. Across all 20 motions in the dataset, the average within-motion alignment is $0.99998 \pm 8.91 \times 10^{-6}$, empirically confirming the effectiveness of the alignment objective.

In practice, overlapping states across different motions can make the motion embeddings harder to distinguish. We mitigate this issue by introducing a phase variable $t$ and concatenating it to each frame $o_t$, as well as defining each state as a stack of the most recent $n$ frames, $s_t = [o_{t-n+1}, o_{t-n+2}, \ldots, o_t]$. This reduces the likelihood that two motions share identical frame sequences at the same phase $t$. In our dataset, $n = 5$ works well, though the optimal value may vary depending on the data. Additionally, our method is robust to a small number of overlapping states, as long as they do not significantly pull the motion-level embeddings of different motions closer together.

## F    FULL ALGORITHM OF RGSD

---

**Algorithm 1** RGSD - Pretraining

---

1: Initialize encoder function $q_\phi(s)$, reference dataset $\mathcal{M}$
2: **for** $i \leftarrow 1$ to # of epochs **do**
3:     Sample $m \sim \mathrm{Uniform}(\mathcal{M})$
4:     Sample $x^a, x^+$ from $m$, $x^-$ from $\{\mathcal{M} - m\}$
5:     Update $q_\phi$ with $\mathcal{L}_{\mathrm{InfoNCE}}$ on Eq. 8
6: **end for**

---

After pretraining is completed, imitatino and discovery happen in parallel.

---

**Algorithm 2** RGSD: Imitation & Discovery

---

1: **Initialize:** skill-conditioned policy $\pi_\theta$, frozen pretrained encoder $q_\phi^-$, trainable encoder $q_\phi$, imitation ratio $p$, KL loss coefficient $\alpha$, reference dataset $\mathcal{M}$, set of reference motion embeddings $\mathcal{Z}_{\mathrm{ref}}$, feature mask $\phi_{\mathrm{mask}}$, and replay buffer $\mathcal{D}$
2: Initialize $q_\phi \leftarrow q_\phi^-$
3: **for** $i \leftarrow 1$ **to** number of epochs **do**
4:     **for** $j \leftarrow 1$ **to** episodes per epoch **do**
5:         Sample task indicator $c \sim \mathrm{Bernoulli}(p)$          // 1 being imitation, 0 being discovery
6:         Sample reference $m \sim \mathrm{Uniform}(\mathcal{M})$
7:         Initialize $s \sim \mathrm{Uniform}(m)$
8:         Compute $z_m$ using Eq. 5 with $q_\phi^-$.
9:         $z \leftarrow z_m \cdot c + k/\|k\| \cdot (1 - c)$, where $k \sim \mathcal{N}(0, I)$
10:
11:         **while** episode not done **do**
12:             Execute $a \sim \pi_\theta(\cdot \mid s, z)$, observe $s'$
13:             Compute reward $r \leftarrow \mathrm{Const.} + \log q_\phi^-(z \mid s') \cdot c + \log q_\phi(z \mid s') \cdot (1 - c)$
14:             Add $\{s, a, r, s', z, c\}$ to $\mathcal{D}$
15:             $s \leftarrow s'$
16:         **end while**
17:     **end for**
18:     **for** each $\{s, a, r, s', z, c\} \in \mathcal{D}$ **do**
19:         Compute $\mathcal{L}_\phi \leftarrow (1 - c) \cdot (-\log q_\phi(z \mid s')) + \alpha * c \cdot \mathrm{KL}\big(q_\phi(\cdot|s) \,\|\, q_\phi^-(\cdot|s)\big)$
20:         Update $\phi$ to minimize $\mathcal{L}_\phi$
21:         Update $\theta$ using PPO with rewards $r$
22:     **end for**
23: **end for**

---

## G    EXPERIMENTS DETAILS

### G.1    STATE AND ACTION SPACE

We use a 3D SMPL humanoid character, commonly adopted in prior work (Luo et al., 2023; Tessler et al., 2024; Tirinzoni et al., 2025). The state representation is defined as follows:

- phase variable: 1-D
- root (pelvis) height - 1-D
- root rotation relative to the local coordinate frame, 69-D
- local rotation of each joint, , 144-D
- local velocity of each joint, and , 72-D
- positions of the hands and feet in the local coordinate frame. 72D

While policy and value function takes single frame observation as input, encoder takes 5 frames concatenated the latest 5 steps of observations as input, $s_t = (o_{t-4}, o_{t-3}, o_{t-2}, o_{t-1}, o_t)$. The agent controls the character by outputting target rotations for PD controllers at each joint. In total the character has 23 spherical joints, resulting in 69D action space.

### G.2 IMPLEMENTATION DETAILS OF BASELINE ALGORITHMS

To isolate the algorithmic contributions, we removed all additional engineering techniques applied in the original implementations to improve motion robustness.

For **ASE**, the original training introduces recovery strategies by initializing $10\%$ of the environments in random fallen states. While this clearly improves robustness, we removed this component consistently across our method and all baselines. ASE also incorporates an auxiliary diversity objective:

$$\mathcal{J}_{\text{div}} = \mathbb{E}_{d^\pi(s)} \, \mathbb{E}_{z_1, z_2 \sim p(z)} \left[ \left( \frac{D_{\text{KL}}(\pi(\cdot \mid s, z_1), \, \pi(\cdot \mid s, z_2))}{D_Z(z_1, z_2)} - 1 \right)^2 \right] \tag{13}$$

where $D_Z$ is a distance function. Together with the mutual information objective, $\mathcal{J}_{\text{div}}$ encourages policies to exhibit diverse behaviors when conditioned on different latent variables. Although this objective is modular and could be applied to any baseline, we excluded it to focus strictly on the main algorithmic differences.

For **CALM**, we encoded entire motions rather than segmenting them into 2-second clips, since the motion lengths we used (between 2 and 5 seconds) are relatively short. Shorter motions were zero-padded to standardize sequence lengths.

Finally, following conventions in skill discovery frameworks (Gregor et al., 2016; Eysenbach et al., 2018; Laskin et al., 2022; Park et al., 2023b; Rho et al., 2024), we sampled a skill latent at the beginning of each episode and kept it fixed throughout the rollout. While transitioning between skills is orthogonal and can be incorporated into both our method and the baselines to further improve robustness, we excluded it here to focus on a clean understanding of algorithmic contributions.

### G.3 LIST OF REFERENCE MOTIONS

We list the file names (relative paths) of all motions used in our experiments:

1. `ACCAD-smpl/Male2Walking_c3d/B10____Walk_turn_left_45`
2. `ACCAD-smpl/Male2Walking_c3d/B9____Walk_turn_left_90`
3. `ACCAD-smpl/Male2Walking_c3d/B11____Walk_turn_left_135`
4. `ACCAD-smpl/Male2Walking_c3d/B14____Walk_turn_right_45_t2`
5. `ACCAD-smpl/Male2Walking_c3d/B13____Walk_turn_right_90`
6. `ACCAD-smpl/Female1Walking_c3d/B22___side_step_left`
7. `ACCAD-smpl/Female1Walking_c3d/B23___side_step_right`
8. `ACCAD-smpl/Male2Running_c3d/C12___run_turn_left_45`
9. `ACCAD-smpl/Male2Running_c3d/C11___run_turn_left_90`
10. `ACCAD-smpl/Male2Running_c3d/C13___run_turn_left_135`
11. `ACCAD-smpl/Male2Running_c3d/C15___run_turn_right_45`
12. `ACCAD-smpl/Male2Running_c3d/C14___run_turn_right_90`
13. `ACCAD-smpl/Male2Running_c3d/C16___run_turn_right_135`
14. `ACCAD-smpl/Male2Running_c3d/C7___run_backwards_t2`
15. `ACCAD-smpl/Female1Walking_c3d/B5___walk_backwards`
16. `ACCAD-smpl/Male2Walking_c3d/B5____Walk_backwards`
17. `ACCAD-smpl/Male2MartialArtsPunches_c3d/E3____cross_left`
18. `ACCAD-smpl/Male2MartialArtsPunches_c3d/E4____cross_right`
19. `ACCAD-smpl/Male2MartialArtsPunches_c3d/E7___uppercut_left`
20. `ACCAD-smpl/Male2MartialArtsPunches_c3d/E8___uppercut_right`

## G.4 HYPERPARAMETERS

Table 2: Hyperparameters of RGSD

| Name | Value |
| --- | --- |
| **Network Architecture** | |
| Dim. of latent $z$ | 16 |
| Policy network $\pi$ | MLP with [1024, 1024, 1024, 512], |
| Activaion of $\pi$ | tanh |
| Encoder network $q_\phi$ | MLP with [1024, 1024, 1024, 512] |
| Activaion of $q_\phi$ | ReLU |
| $q_\phi$ input frame stacks | 5 |
| **Contrastive pretraining** | |
| Minibatch size | 256 |
| Total number of epochs | 3,000 |
| Optimizer | Adam(Kingma & Ba, 2014) |
| Learning rate for encoder | 1e-4 |
| **Imitation and Discovery** | |
| Ratio of imitation environments $p$ | 0.7 |
| Learning rate | 2e-5(actor), 1e-4(critic), 1e-4(encoder) |
| Optimizer | Adam |
| Minibatch size | 32768 |
| Horizon length | 32 |
| PPO clip threshold | 0.2 |
| PPO number of epochs | 5 |
| GAE $\lambda$ (Schulman et al., 2015) | 0.95 |
| Discount factor $\gamma$ | 0.99 |
| Entropy coefficient | 0.1 |
| KL loss coefficient | 0.5 |

## H SUPPLEMENTARY DISCUSSION: CHALLENGES OF EXPANDING RGSD WITH METRA

In Section 6, we have discussed why the METRA objective conflicts with learning repetitive motions when only local observations are available. A potential workaround is to augment the state with additional variables, such as a global coordinate or time variable. These variables could help distinguish between locally identical motions, e.g., before and after taking a step from walking motion. However, each approach ultimately faces a challenge for different reasons.

**Global coordinate information.** Given that all state information is computed in the agent's local coordinate system, the first and last states of a walking behavior appear identical. Concatenating global coordinate to the state would allow the state to capture this progress. However, this approach still cast issues. First, not all motions produce meaningful displacement in global coordinates (e.g., shaking hands). More importantly, incorporating global position creates an imbalance in the state representation. All other state dimensions are bounded, which implies that the output of $\phi$ is bounded as well. In contrast, global position is unbounded, and METRA can exploit this property. Recall that the METRA reward is defined as $[\phi(s') - \phi(s)]^\top z$, where reward increases when the agent visits new states $s'$ along the $z$ direction in latent space. Because global position is unbounded, changes in position always yield new states, regardless of behavior. Consequently, the agent ends up learning skills that simply move in different directions when conditioned on different $z$, since this suffices to maximize the reward.

**Time variable.** A time variable could serve as a meaningful alternative to the global coordinate, as it is less likely to be exploited by the METRA reward. Although the time variable is also unbounded, it advances uniformly across all behaviors regardless of the underlying dynamics, it increments by

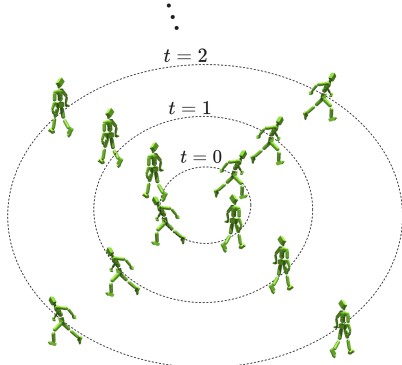

Figure 8: Conceptual figure of latent space learned by METRA with time variable.

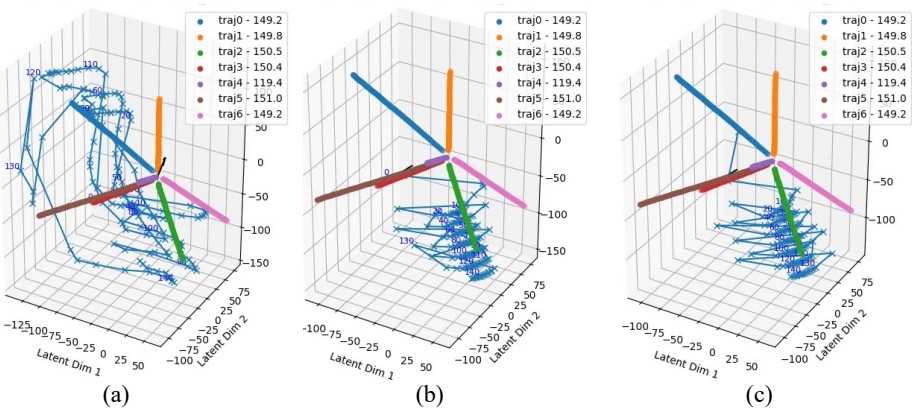

Figure 9: Latent space learned with **METRA**. It shows seven reference trajectories and the agent's trajectory (blue line with x markers). We introduce a time variable to differentiate locally identical state pairs in repetitive motions, which causes the agent trajectory to exhibit discontinuous transitions in the latent space.

a constant amount at every step. This ensures that no behavior can accelerate or decelerate its progression. Moreover, it effectively differentiates identical states over time. Therefore the agent retains the opportunity to learn meaningful behaviors, rather than exploiting state changes of specific dimensions.

However, this design still introduces issues. Figure 8 illustrates an ideal latent space of reference motions formed by METRA with a time variable. Each motion is aligned with a distinct directional vector, and consecutive states are placed at a distance of 1. In this setup, the time variable naturally induces a *contour*: distances in the latent space correspond to "temporal distances," i.e., the minimum number of transition steps required to reach a given state. For example, if the time variable is 3, then reaching that state requires exactly three transitions, and at the optimum of the representation function $\phi$, all behaviors at timestep 3 should lie on the same contour.

The core problem arises during METRA exploration. When the agent transitions at large $t$ values—for example, from the $t=100$ contour to the $t=101$ contour—it may undergo an *abrupt transition* of considerable magnitude in the latent space, especially if the subsequent pose lies on the opposite side of the contour. This phenomenon is illustrated in Fig. 9. Since METRA enforces that the latent distance between successive states must be less than 1, such discontinuities break the contour structure. As a result, temporal coherence in the latent space is lost, and the representation collapses. In short, constructing an idealized latent space from reference data while respecting ME-TRA constraints with a phase variable introduces a fundamental conflict, leading to highly unstable training.

