# OpenReview forum: "Reference Grounded Skill Discovery"
_ICLR.cc/2026/Conference — ICLR 2026 Poster_

### Official Review · Reviewer_1tD4 · 2025-10-24

**Soundness:** 3
**Presentation:** 3
**Contribution:** 2
**Rating:** 4
**Confidence:** 4

**Summary:**

This paper proposes a way to do unsupervised skill discovery based on reference motion data. The authors first learn a latent mapping function (from $\mathcal{S}$ to $\mathcal{Z}$) with contrastive learning. In the next phase, they train a DIAYN-like objective with a KL constraint to train a skill policy. As a result, they get a set of skills that are diverse while looking natural. They compare their method with two unsupervised skill discovery baselines (DIAYN and METRA) and two imitation learning baselines (ASE and CALM) in terms of $\ell^2$ distance and FID, showing that their method produces better behaviors.

**Strengths:**

The paper is well-motivated and generally well-written. The problem setting in this paper (i.e., how to do skill discovery based on some prior knowledge -- in this case, a reference motion dataset) is important and timely. While the individual components of the proposed method are not necessarily novel, the combination of these components has (to my knowledge) not been previously done in the literature. I also like that the proposed method is straightforward and relatively easy to implement. The authors convincingly demonstrate that RGSD is more effective than ASE and CALM.

**Weaknesses:**

The main weakness of this work is in the empirical comparison. While the authors compare their method against ASE and CALM, there are many more relevant previous works (essentially the whole literature on "data-driven/offline skill discovery") that are closely related to the proposed method but are not properly discussed/compared against. For example, OPAL and SPiRL also do the conceptually similar thing as this method -- they first learn a latent space (via a VAE) and then do policy learning based on the learned latent space. I know these methods are not exactly the same as the proposed one (and the problem settings are also not identical), but I think they are still close enough, to the degree that I would have expected to see comparisons with prior works in data-driven skill discovery. I also would have expected more extensive discussions of this line of research.

The paper distinguishes its method from Motivo by saying that "it does not explicitly aim to discover new skills, which distinguishes our contribution," which I find unconvincing. I agree that the latent space is not densely covered by $z$'s from the dataset trajectories, but given that the distribution of latent vectors tends to be uniform with contrastive learning (https://proceedings.mlr.press/v119/wang20k/wang20k.pdf), interpolated latents are still more or less in-distribution. Moreover, even if one treats these interpolated vectors as "new skills," the same argument could apply to other works including Motivo, FB, etc. as well, where one samples $z$'s from the prior distribution (typically a (unit) Gaussian) to train a policy. Hence, I think the authors should either provide empirical quantitative evidence that shows RGSD indeed learns more novel skills, or tone down the claim about the ability to discover "new" skills. I also think the paper could have been much stronger with comparisons with Motivo, as it is one of the most closely related works both algorithmically and empirically.

I'd be happy to adjust my score if the authors address these points.

**Questions:**

I don't have particular questions about this paper as of now. My main issue is the lack of discussion and empirical comparison with related work (see the weaknesses section above), which I hope the authors can address during the rebuttal period.

---

> ### Author Response · Authors · 2025-11-29
>
> Thank you for your valuable comment. Based on your suggestion, we incorporated Meta-Motivo as an additional baseline in all of our experiments. I understand that you may no longer be able to update your review due to the circumstances, but I still wanted to share the progress we made. All experiments have been updated in the revised PDF, and we kindly refer you to that version for full details.
>
> **Summary of Updates**
>
> - **Imitation capability (Section 5.1).**
> Meta-Motivo achieves the best FID scores on 4 out of 5 motions. In contrast, RGSD (ours) achieves the lowest Cartesian tracking error on 4 out of 5 motions, suggesting that while RGSD may produce slightly less natural motions, it excels in high-fidelity and temporally aligned reproduction of reference motions.
>
> - **Discovered skills (Section 5.2):**
> Meta-Motivo continues to produce the most natural variations of the original motions, achieving the best FID scores and generating trajectories that remain well-centered around the reference motion embeddings.
>
> - **Downstream task performance (Section 5.3):**
> For high-level policy training, only RGSD was able to consistently reach randomly spawned goals while respecting the commanded style, while others failed to fully respecting the directed style. In the freestyle task, RGSD was also the most sample-efficient method, reaching a 100% success rate with the fewest samples. Additional explanations and analyses are provided in the updated paper.
>
> We sincerely appreciate your suggestion—our work has become much stronger thanks to your feedback.

---

### Official Review · Reviewer_Yhxb · 2025-10-31

**Soundness:** 3
**Presentation:** 3
**Contribution:** 3
**Rating:** 8
**Confidence:** 3

**Summary:**

The paper introduces Reference-Guided Skill Discovery (RGSD), a framework for reproducing and controlling diverse motions using reference data, outperforming prior imitation and unsupervised skill discovery methods. RGSD first learns a latent skill space by contrastively embedding reference trajectories, clustering each motion around a unique latent direction. It then jointly trains a skill-conditioned policy to imitate reference motions and discover novel skills by conditioning on both reference and interpolated latents. In experiments, RGSD consistently outperforms prior approaches by achieving lower Cartesian errors and maintaining high trajectory fidelity across reference trajectories, demonstrating strong improvements over both imitation-based and unsupervised skill discovery baselines in high-DoF SMPL humanoid.

**Strengths:**

**Strengths:**

- Clear, easy to read
- Demonstrates strong, consistent improvements over established baselines
- Provides a principled advancement over both imitation-based and unsupervised skill discovery approaches

**Weaknesses:**

**Weaknesses:**

- Downstream task evaluation is limited to a single, relatively simple task
- Assumes coverage of skill in the pretraining reference dataset
- Limited exploration of performance with diverse or noisy reference data

**Questions:**

- How does RGSD perform when reference trajectories are noisy or highly variable, as in real-world human reference trajectories due to inaccuracies in sensing or mapping?

- Can RGSD effectively scale and generalize as the number and diversity of reference trajectories increase?

- Is the set of discovered skills strictly limited to local variants of those present in the reference dataset? If yes, I would suggest clarifying this in the paper

- How does RGSD compare to frameworks like Meta-Motivo, which also address a similar problem of acquiring skills from a reference dataset, despite not focusing exclusively on skill discovery?

- How does RGSD offer advantages over CALM? Both methods achieve similar downstream task performance. Are there settings where RGSD’s skill interpolation provides unique benefits?

---

### Official Review · Reviewer_8xWQ · 2025-11-01

**Soundness:** 2
**Presentation:** 3
**Contribution:** 3
**Rating:** 4
**Confidence:** 3

**Summary:**

* The main contribution of this paper is a framework that improves the initialization of skill discovery by using a seed set of reference motions that the discovery algorithm first imitates, followed by an unsupervised phase that introduces novelty and diversity without straying too far from the reference behaviors. The approach relies on constructing and exploiting a semantically grounded latent space, obtained through a contrastively pretrained encoder that maps states onto a unit-sphere latent representation via a von Mises–Fisher (vMF) distribution.
In practice, the imitation and discovery processes are performed in parallel: the pretrained encoder is frozen for reward computation, while a second encoder is trained through reinforcement learning to track it under a KL-divergence penalty.

* Experimentally, on a 69-DoF SMPL humanoid with a 359-dimensional observation space, the proposed method successfully imitates the reference motions (walk, run, sidestep, backward, and punch) with low Cartesian error and achieves competitive FID metrics compared to DIAYN, METRA, ASE, and CALM. For the downstream “GoalReaching-Sidestepping” task, a high-level policy based on RGSD attains a success rate comparable to CALM, while achieving a better FID score.

**Strengths:**

* The paper is clearly written and well structured. The proposed method is well motivated and exhibits an elegant design, both in its overall pipeline and in the chosen geometry of the latent space.
* The experimental section provides clear and convincing qualitative evidence supporting the effectiveness of the approach.

**Weaknesses:**

* Several claims in the paper appear overstated. In particular, the theoretical analysis relies on the assumption of perfect within-motion alignment, which is unlikely to hold in practice. The encoder has finite capacity, and although the InfoNCE objective promotes alignment, it does not guarantee it. Moreover, the authors introduce several heuristic mechanisms—such as artificially terminating episodes when deviations from alignment become too large—to make the method work in practice. Unfortunately, all theoretical developments are built upon this idealized assumption, and no convergence or stability bounds are provided to quantify the impact of suboptimal alignment.

* The comparisons with baselines are also somewhat unfair. For example, ASE and CALM were originally proposed with explicit diversity components, yet these elements were omitted in the reported baseline implementations. This undermines the strength of the comparative results.

* The paper further relies heavily on the Fréchet Inception Distance (FID) as a key evaluation metric, but the justification for its use in this context is insufficient. Moreover, visual inspection of the figures suggests that small FID differences sometimes contradict qualitative impressions, casting doubt on its reliability as the primary measure of performance.

* Finally, the choice of downstream task appears somewhat ad hoc. The corresponding reward function is explicitly designed to favor a latent skill close to the “sidestep” motion encoding, introducing a bias that aligns the task with the model’s strengths. In more general settings, downstream tasks will not necessarily be directional in this sense. The paper implicitly assumes that “motions” correspond directly to “directions” in latent space, which limits the generality of the approach.

**Questions:**

See suggestions of improvement in the "Weaknesses"  Section.

---

> ### Author Response · Authors · 2025-11-22
>
> We appreciate your review. Below, we provide a topic-wise response to address your concerns.
>
>
> **[w1] Assumption of perfect within-motion alignment**
>
> We agree with the reviewer’s concern regarding within-motion alignment. Though we have provided theoretical optimum for the InfoNCE loss, we agree that providing the empirical evidence would be important. To empirically verify this observation, we conducted an additional analysis measuring the degree of within-motion alignment. According to our experiments, we consistently observed that a neural network can easily fit to the data, producing embeddings with nearly perfect within-motion alignment.
>
> In detail, we first visualized the latent embeddings of each motion in a 3D space using the pretrained encoder $\mu_\phi$. Since the latent space is 16-dimensional, we show only the first three dimensions. The resulting plots are newly included in Appendix D. As shown in the figure, after roughly 100 epochs, each motion forms a tight cluster, indicating near-perfect alignment.
>
> We also quantify this alignment using cosine similarity. For each motion $m_i \in \mathcal{M}$, we compute its motion-level embedding $z_{m_i}$. Then, for every state $s \in m_i$, we measure the cosine similarity between its latent embedding and $z_{m_i}$, and average these values to obtain the within-motion alignment score. Across all 20 motions in the dataset, the average within-motion alignment is $0.99998 \pm 8.91 \times 10^{-6}$, confirming the effectiveness of the alignment objective.
>
> Finally, we note that while overfitting $\mu_{\phi}$ may produce a ragged latent space for in-between motion embeddings, this does not affect the discovery phase. As described in Section 4.3, the original $\mu_{\phi}$ is used only (1) to initialize the parameters of $\mu_{\phi}'$ and (2) to compute embeddings for the reference motions. During discovery, $\mu_{\phi}'$ is updated online, and the in-between latent space is gradually shaped by newly gathered agent trajectories, mitigating any irregularities inherited from the pretrained encoder.
>
>
>
> **[w2] Potential unfairness for baseline comparison with ASE and CALM**
>
> We acknowledge that we intentionally excluded the extra diversity loss in Eq. (13) for **ASE**, as stated in Section F.3. However, our intention behind removing this additional loss was to ensure a fair comparison, not to introduce any bias in favor of our approach.
>
> It is important to note that, due to the extra diversity loss, ASE originally includes **two sources of diversity**: one from mutual information maximization and another from the additional diversity loss. In contrast, RGSD relies **only** on mutual information maximization. By removing the extra diversity term from ASE, both methods rely on the same diversity mechanism. The only remaining difference lies in the imitation objective: ASE uses a GAIL-based objective, whereas RGSD employs our proposed reference grounding mechanism. We believe this setup provides a clearer and more controlled analysis of the impact of reference grounding.
>
> Another point is that the extra diversity loss in ASE is **orthogonal** to the method itself, meaning that it could also be added to RGSD if desired.
>
> That said, we understand the reviewer’s concern that omitting this loss might give a misleading impression of ASE’s performance. To address this, we retrained ASE from scratch using the 20 motions while including the extra diversity loss. Our findings show that ASE with the extra loss still suffers from the same issue as the original ASE, namely **mode collapse**, where the policy struggles to distinguish between very similar motions within the same category.
>
> Below are the detailed results. Values represent averages over 500 episodes per motion, aggregated by category.
>
> | **Cartesian err** | Walk | Run | Sidestep | Backward | Punch |
> |-------------------|------|------|----------|----------|--------|
> | ASE without diversity | 8.2 | 16.4 | 10.3 | 11.6 | 9.0 |
> | ASE with diversity | 11.1 | 16.8 | 12.4 | 13.0 | 11.4 |
>
> | **FID score**     | Walk | Run | Sidestep | Backward | Punch |
> |-------------------|------|------|----------|----------|--------|
> | ASE without diversity | 2.9 | 11.3 | 22.2 | 62.6 | 4.9 |
> | ASE with diversity | 11.0 | 11.6 | 19.6 | 61.0 | 4.7 |
>
> As shown in the tables, the extra diversity loss has limited impact on mitigating mode collapse. CALM partially resolves this issue through its conditional discriminator, which contributes to its improved results.
>
> Regarding **CALM**, we could not fully identify the explicit diversity components mentioned by the reviewer. If you could point us to a specific section or equation, that would be very helpful. If you were referring to the uniformity loss of the motion encoder in Eq. (6) of the CALM paper, that component is already included in our CALM experiments. If you meant something else, please clarify, and we would be happy to incorporate it.

---

> ### Author Response · Authors · 2025-11-22
>
> **[w3] Justification of FID**
>
> We understand the concern regarding the use of the FID score. To demonstrate its relevance to generated motion quality, we extensively recorded the output motions produced by RGSD as well as by baseline methods, and we report their corresponding FID values.
>
> In **video-part2**, we present videos and FID scores for RGSD on four different motions, each evaluated under three levels of diversity. This video illustrates that RGSD can control the degree of diversity at test time by adjusting the latent sampling distribution. Since increased diversity often comes at the cost of motion naturalness, the generated motions become less realistic compared to the reference motions visualized in **video-part0**. As performance degrades, the **FID scores consistently increase** across all motions, showing that FID provides a reasonable proxy for the similarity between reference and generated motions.
>
> In **video-part5**, we show the behaviors of RGSD and CALM trained on downstream tasks along with their FID scores. When the trained policy successfully follows the commanded style, such as sidestepping or backward movement, the corresponding FID score is low. When the policy ignores the commanded style and instead defaults to forward walking, the FID score becomes high. This again highlights the reliability of using FID as an evaluation metric.
>
> Lastly, we note that the FID score is widely used in prior work as a quantitative metric for motion naturalness and similarity, including the following papers:
>
> 1. Scene-Aware Generative Network for Human Motion Synthesis (Wang et al., 2021)
> 2. Human Motion Diffusion Model (Tevet et al., 2022)
> 3. InsActor: Instruction-Driven Physics-Based Characters (Ren et al., 2023)
> 4. MotionGPT: Human Motion Synthesis with Improved Diversity and Realism via LLMs (Ribeiro-Gomes et al., 2024)
> 5. Motion In-Betweening for Densely Interacting Characters (Zhang et al., 2025)
>
>
> **[w4] The choice of downstream task**
>
>
> Please note that the reward setting we used for the downstream task is identical to that of **CALM (Tessler et al., 2023)**. The goal of the downstream task is twofold:
> - reaching a randomly spawned goal,
> - with a designated movement style.
>
> Accordingly, the reward design presented in Section 5.3 is the sum of two components, $r_{task}$ and $r_{style}$, where $r_{style}$ encourages the high-level policy $\pi_{high}$ to output latent action $z$ that are close to the intended motion embedding $z_m$.
>
> That said, we understand that the scope of the downstream task may appear limited. To address this, we expanded the **GoalReaching** tasks to include three distinct styles:
> - **Freestyle**
> - **Sidestepping**
> - **Backward**
>
> In each episode, the agent must reach the goal while adhering to the specified style. For example, in the Sidestepping and Backward tasks, the agent must reach the target by sidestepping or moving backward. In the **Freestyle** task, the agent is not constrained to a particular style and may select skills freely.
>
> These tasks are inherently challenging because the reference motions for sidestepping and backward running do not include any turning behaviors. Thus, reaching goals at diverse positions while respecting the style requires discovering additional skills, such as backward turns at various angles. Imitation-based baselines lack this capability, whereas RGSD acquires diverse turning behaviors and is therefore able to reach the goal while maintaining the intended style.
>
> We provide qualitative comparisons for both RGSD and CALM in **video-part5**. For **RGSD**, the agent successfully learned to reach the goal while preserving the commanded style. Notably, in the **Backward** task, when the goal appeared directly in front of the agent, the agent chose to take a detour so that it could still approach the goal while moving backward, demonstrating strong adherence to the stylistic constraint.
>
> In contrast, **CALM** reliably reached the goal but failed to maintain the designated style: the agent defaulted to forward walking regardless of the commanded style. This discrepancy is also reflected in the FID scores shown in the video. For example, RGSD achieved an FID score of 6.8 on the sidestepping task, whereas CALM’s score was 29.3.
>
> We have incorporated these results into the updated paper.

---

### Official Review · Reviewer_a8F6 · 2025-11-06

**Soundness:** 2
**Presentation:** 3
**Contribution:** 2
**Rating:** 2
**Confidence:** 4

**Summary:**

This paper introduces Reference-Guided Skill Discovery (RGSD), a method that enables unsupervised skill discovery to scale to high degree-of-freedom robots by grounding the exploration process in a semantically meaningful latent space derived from reference motion data. The key insight is to first use contrastive learning to embed reference motions onto a unit hypersphere, then leverage this pre-structured latent space to simultaneously imitate reference behaviors (by sampling latent vectors along reference directions) and discover novel related skills (by sampling novel directions). Applied to the SMPL humanoid, RGSD successfully learns both to reproduce complex motions like walking, running, and punching with high fidelity, and to discover semantically coherent variations of these behaviors-addressing the fundamental challenge that pure exploration in high-dimensional spaces tends to produce meaningless, unstructured behaviors while pure imitation fails to generalize beyond the reference dataset.

**Strengths:**

The paper provides rigorous theoretical justification for its core technical claims (Sec.A, Sec.B, Sec.C.1, etc.).

**Weaknesses:**

- **Unclear scope of "discovery" and potential overselling.** The paper's motivation (first version, lines 44-53) suggests learning skills beyond the reference data that are still "semantically meaningful," using the example of a manipulator learning diverse skills like pushing and grasping. However, the experimental results demonstrate a more limited form of discovery: variations of existing reference behaviors (e.g., punching in different directions, sidestepping with varying turns) rather than genuinely novel skill categories. While this is still valuable, it conflicts with the initial framing. The paper would benefit from: (a) explicitly clarifying that discovered skills remain within the semantic manifold spanned by references, or (b) experiments sampling latent vectors farther from reference embeddings to empirically characterize what "novel" behaviors emerge and whether they remain meaningful or degenerate into unstructured motion.

- **Strong assumptions on data segmentation limit practical applicability.** The method assumes each reference motion corresponds to a single, consistent skill throughout its duration. While this holds for the curated dataset used in experiments, most real-world video or robot demonstration data contains multiple skills within single trajectories and would require extensive manual segmentation and annotation. The paper provides no discussion of: (a) how to handle multi-skill demonstrations, (b) robustness to segmentation errors, or (c) potential extensions to learn from unsegmented data. This significantly limits the method's applicability beyond carefully curated motion capture datasets.

- **Insufficient downstream task evaluation.** Sec.5.3 presents only a single downstream task (goal-reaching with sidestepping), which is insufficient to validate the learned skills' general utility. Additionally, critical experimental details are missing, e.g., how to obtain the frozen $\pi_\text{low}$

- No supplementary material (e.g., videos of rollout examples) is provided.

**Questions:**

- **Validation of within-motion alignment.** While Appendix B proves that contrastive pretraining achieves within-motion alignment at optimality, can you provide empirical evidence that this property is achieved in practice? Specifically: (a) quantitative metrics showing the variance of embeddings $\mu_\phi(s)$ for states $s$ within the same trajectory, (b) visualization of how embeddings cluster on the unit hypersphere for different motions, and (c) analysis of whether this alignment quality correlates with downstream imitation performance. Additionally, does this alignment property degrade when sampling latent vectors far from reference embeddings during discovery?

- **Sensitivity to reference dataset composition.** The experiments use a fixed set of 20 reference motions (first version, lines 336-342). How does the method's performance scale with: (a) the number of reference motions per category (e.g., 1 vs. 5 walking motions), (b) the total number of skill categories, and (c) the diversity within each category (e.g., straight walking vs. walking with various turns)?

- **Qualitative comparison on downstream task behaviors.** While Sec.5.3 reports FID scores showing RGSD (34.3) outperforms CALM (46.7) on the goal-reaching task, could you provide visual comparisons showing the actual behavioral differences? Given that both methods achieve similar success rates, understanding how they differ qualitatively would better support the claim of learning "semantically meaningful" skills.

**Details Of Ethics Concerns:**

No ethics review is needed since all experiments are conducted in simulations.

---

> ### Author Response · Authors · 2025-11-21
>
> We appreciate your thoughtful review. After reading your comments, we realized that the current draft would benefit from stronger empirical evidence to support the effectiveness of our method. In response, we conducted a series of additional experiments and obtained positive results across multiple aspects. Please refer to the **attached video** for the full set of new experimental findings.
>
> **[w1] Unclear scope of "discovery" and potential overselling**
>
> We agree with your concerns. To better understand the degree of novelty for skills discovered by **RGSD**, we conducted two additional experiments.
>
> **First**, we examined how sampling latent vectors farther from the motion embedding affects behavior. Our hypothesis was that this would increase behavioral diversity, potentially at the cost of motion naturalness. Concretely, since our sampling uses a von Mises–Fisher (vMF) distribution, we progressively decreased the concentration parameter from 1,000 → 100 → 20 while keeping the mean direction fixed to each motion’s embedding. We repeated this analysis on four motions: side-stepping, run-turn-left-135, punch-left-cross, and run-backwards.
>
>  As shown in **video-part2**, reducing the concentration parameter indeed produces more diverse behaviors, with only limited degradation in motion quality. For instance, when starting from sidestepping-right motions, broader sampling causes the resulting trajectories to spread more widely. As illustrated in the video, such controlled diversity is not commonly observed in imitation-based baselines such as CALM or META-MOTIVO.
>
> **Second**, we pushed this further by sampling each episode’s latent vector uniformly from the entire hypersphere. The resulting behaviors are shown in **video-part4**. In this setting, we observed genuinely novel behaviors—such as various turning motions, distinct standing poses, different running styles, clap-like motions, and several compositional behaviors (e.g., run → shake, walk → pose, run → turn). However, we also observed corresponding decreases in motion naturalness or structuredness. Also we agree that although these behaviors are something novel, they still remain within the semantic manifold spanned by references.
>
> We plan to incorporate these findings into the revised version of the paper. We also agree that referencing manipulation skills could be misleading, so we have removed that phrasing. If there are any additional statements that feel overstated, please let us know—we are happy to revise them further.
>
>
> **[w2] Strong assumptions on data segmentation**
>
> Yes, the current set of 20 motions consists of single-skill demonstrations, although some motions such as *run_turn_left_135* could be interpreted as a composition of *run-forward* followed by *turn-left*.
>
> Following your advice, we conducted an additional experiment using **multi-skill reference** motions. From the ACCAD dataset, we selected 14 multi-skill motions and trained RGSD from scratch. Some examples include:
>
> - run_backwards_stop_run_forward
> - run_to_crouch
> - walk_to_pickup_box
> - walk_to_skip
>
> The full list and resulting behaviors are provided in **video-part3**.
>
> We observed that RGSD still learns meaningful variations of these multi-skill demonstrations. For instance, with the *run_backwards_stop_run_forward* motion, agents begin by running backward, but then stop at diverse locations and continue running forward in various directions.
>
> However, we still believe that RGSD is most effective when trained on single-skill reference motions. Multi-skill references tend to be tailored to a narrow range of tasks, whereas primitive skills can be composed to solve a much wider variety of tasks.

---

> ### Author Response · Authors · 2025-11-21
>
> **[w3, q3] Insufficient downstream task evaluation & qualitative comparison on downstream task training**
>
> We agree that the downstream evaluation in the original draft was limited. To address this, we expanded the **GoalReaching** tasks to include three distinct styles:
>
> 1. **Freestyle**,
> 2. **Sidestepping**, and
> 3. **Backward**.
>
> In each episode, the target goal is randomly placed within a 20 m × 20 m square, and the agent must reach the goal while adhering to the designated style—for example, by sidestepping or moving backward. In the **Freestyle** task, the agent is not constrained to a particular style and may choose skills freely.
>
> We evaluate performance using two key metrics:
>
> - **Success rate**, and
> - **FID score**, which measures how closely the agent’s motion matches the corresponding style-specific reference dataset.
>
> Qualitative results for both RGSD and CALM are shown in **video-part5**.
>
>  For **RGSD**, the agent successfully learned to reach the goal while maintaining the specified style. Notably, in the **Backward** task, when the goal appeared directly in front of the agent, the agent chose to take a detour so that it could reach the goal while still moving backward, demonstrating strong adherence to stylistic constraints.
>
> In contrast, **CALM** reached the goal reliably but failed to maintain style: the agent moved forward regardless of the designated style. We believe this is an expected outcome, because achieving backward goal-reaching requires multiple backward “turn” variants, but since the reference dataset contains no backward turning motions, CALM is unable to discover such skills. RGSD, however, successfully discovers these turning variants and leverages them to complete the task.
>
> This difference in style adherence is also reflected in the FID scores presented in the video. For instance, RGSD achieved an FID score of 6.8 on the sidestepping task, whereas CALM’s score was 29.3. We computed the FID by comparing 500 agent rollouts per task against the corresponding reference motions. For the freestyle task, where all methods eventually converge to forward walking or running, the FID was computed against the full set of walking and running motions. As a result, RGSD consistently achieves lower FID scores, indicating better alignment with the intended style.
>
> We have incorporated these results into the revised version of the paper.
>
>
> **[w4] No supplementary material**
>
> As mentioned above, we added video results with diverse experiments for rebuttal.
>
>
> **[q1] Validation of within-motion alignment.**
>
> We agree with the reviewer’s concern regarding within-motion alignment. However, in our experiments, we consistently observed that a neural network can easily overfit to the data, producing embeddings with nearly perfect within-motion alignment. To empirically verify this observation, we conducted an additional analysis measuring the degree of within-motion alignment.
>
> Following the reviewer’s suggestion, we first visualized the latent embeddings of each motion in a 3D space using the pretrained encoder $\mu_\phi$. Since the latent space is 16-dimensional, we show only the first three dimensions. The resulting plots are included in **Appendix D**. As shown in the figure, after roughly 100 epochs, each motion forms a tight cluster, indicating near-perfect alignment.
>
> We also quantify this alignment using cosine similarity. For each motion $m_i \in \mathcal{M}$, we compute its motion-level embedding $z_{m_i}$. Then, for every state $s \in m_i$, we measure the cosine similarity between its latent embedding and $z_{m_i}$, and average these values to obtain the within-motion alignment score. Across all 20 motions in the dataset, the average within-motion alignment is $0.99998 \pm 8.91 \times 10^{-6}$, confirming the effectiveness of the alignment objective.
>
> Finally, we note that while overfitting $\mu_{\phi}$ may produce a ragged latent space for in-between motion embeddings, this does not affect the discovery phase. As described in Section 4.3, the original $\mu_{\phi}$ is used only (1) to initialize the parameters of $\mu_{\phi}'$ and (2) to compute embeddings for the reference motions. During discovery, $\mu_{\phi}'$ is updated online, and the in-between latent space is gradually shaped by newly gathered agent trajectories, mitigating any irregularities inherited from the pretrained encoder.

---

> ### Author Response · Authors · 2025-11-21
>
> **[q2] Sensitivity to reference dataset composition.**
>
> We first want to clarify the term **“category”**. In our work, a category is simply a human-made label and does not correspond to any ground-truth semantic grouping. For example, a run backward motion could be placed under the broader “running” category, and sidestepping right could be grouped under “walking.” In fact, our multi-skill training dataset consists of fourteen distinct motions, and each individual motion could be considered its own category. Although we grouped some motions into categories for convenience when presenting results, these labels are used only for interpretation and never used during training.
>
> (a, c) With that clarification, we believe that adding diverse motions within the same category can help expand the set of downstream tasks the agent can solve. Using your phrase, it effectively enlarges the semantic manifold spanned by the reference data. For example, our dataset contains many forward moving motions, including six running and five walking sequences. This diversity enables the agent to perform a wide range of forward behaviors with different speeds and turning angles. We believe this is one of the reasons why, in the freestyle downstream task, all methods ultimately converged to forward walking or running.
>
> (b) Since the notion of category is only conceptual, what matters most is the number and diversity of reference motions, and how large this set can grow while RGSD remains stable. To examine this, we trained RGSD on twenty original motions plus fourteen multi-skill motions, for a total of thirty four motions, without modifying any hyperparameters. Training proceeded smoothly, and RGSD was able to consume this larger collection and reproduce variations of the behaviors, although training took slightly longer(it took about two days with 1 GPU). We are not entirely sure what would happen if RGSD were trained with extremely large collections such as one thousand motions, and we regard this as an interesting direction for future work.

---

> > ### Comment · Reviewer_a8F6 · 2025-11-25
> >
> > Thank you for your response. I have the following follow-up questions and concerns:
> >
> > **Regarding W2 (Compositional Motion Primitives):**
> >
> > My concern is whether RGSD can **disentangle and extract motion primitives that are shared across multiple demonstrations**. For example, in your W2 multi-skill setting, several demonstrations include "walking" as a sub-component. I would like to see evidence that your model can isolate such shared primitives into specific latent codes (without other behaviors). Could you please identify and visualize latent codes that correspond to shared motion primitives (e.g., walking, running, turning)?
> >
> > **Regarding W1 (Semantic Novelty vs. Variation):**
> >
> > This remains my **primary concern**. The examples you provided appear to show variations of existing motions rather than semantically novel behaviors. As an example, you can try to show evidence of **compositional generalization**. For instance, if your training data includes: (1) "run backward → stop running" (2) Multiple demos containing "run forward", and your method can generate "run forward → stop running" by composing these learned primitives, this would demonstrate that your method learns novel skills composed of the original skills.
> >
> > So could you please provide:
> > - Concrete examples of compositionally novel motions that combine sub-behaviors from different training demonstrations in ways not seen during training?
> > - Alternatively, other forms of semantically meaningful novel behaviors that differ qualitatively (not just parametrically) from the training distribution?
> >
> > **Regarding W3 (Comparative Visualization):**
> >
> > Could you please provide:
> > - Side-by-side visualizations comparing CALM and RGSD on the same tasks?
> > - Examples of failure cases for both methods, highlighting when and why each approach breaks down?
> >
> > **Regarding Q1 (RL Fine-tuning Impact):**
> >
> > If I understand correctly, your encoder is further updated during the RL process. Could you please show Figure 6 results using the encoder after RL fine-tuning? This would clarify whether the improved RL performance is partly due to encoder adaptation.
> >
> > **Regarding Q2 (Quantitative Analysis):**
> >
> > Could you please provide:
> > - Quantitative metrics (success rates, motion quality scores, etc.) comparing the different conditions?
> > - Additional qualitative visualizations clearly demonstrating the differences you claim?

---

> > > ### Author Response · Authors · 2025-11-29
> > >
> > > Dear Reviewer a8F6,
> > >
> > > We would like to clarify that the two points you requested evidence for concern capabilities that **we do not claim** in the paper. Specifically, your request asked for:
> > >
> > > 1. Evidence that **RGSD can disentangle and extract motion primitives shared across multiple demonstrations**.
> > > 2. Evidence that **RGSD supports compositional generalization**.
> > >
> > > Regarding the first point, RGSD does **not** attempt to segment demonstrations or extract sub-motion primitives. Our method treats each full demonstration as a **single skill** and discovers semantically related variations around those skills. At no point do we claim that RGSD can decompose motions or isolate shared primitives. In fact, our paper repeatedly and explicitly states that our focus is on discovering semantically similar yet novel variations around reference demonstrations:
> > >
> > > > “RGSD then utilizes this pre-structured latent space to simultaneously imitate reference skills and discover novel behaviors that are semantically related to the references.” (Section 1, Introduction)
> > >
> > > > “Our work can be framed as ‘imitation for discovery,’ because it performs imitation based on reference data to discover semantically similar yet novel behaviors.” (Section 2, Related Works)
> > >
> > > > “RGSD can discover semantically similar yet novel behaviors.” (Section 5.2, Evaluation of Discovered Skills)
> > >
> > > For the second point, compositional generalization is **explicitly identified as future work**, not a capability of the current method. As stated in **Section 7 (Conclusion)**:
> > >
> > > > “One promising direction is to move beyond variants of individual skills toward genuinely compositional behaviors and principled interpolations that blend primitives (e.g., ‘walking while punching’).”
> > >
> > > Given these clarifications, we believe the requested evidence concerns functionalities **outside the stated scope and contributions** of this work.

---

### Author Response · Authors · 2025-12-01

Dear AC,

Thank you for your efforts in handling the review process under challenging circumstances. For your convenience, we summarize the revisions made during the rebuttal period.

---

**1. Expanded downstream task experiments**

The original submission included only a single downstream task experiment. Reviewers 1, 2, and 3 requested additional downstream evaluations. We have now added **two more downstream tasks**, presenting results on **three distinct tasks in total**.
Across all tasks, **RGSD consistently outperforms the baselines**. These updates appear in Section 5.3.

---

**2. Additional baseline: Meta-Motivo**

Reviewers 3 and 4 requested inclusion of **Meta-Motivo** [Tirinzoni et al., 2025] as a baseline. We have added Meta-Motivo to **all experiments** in Sections 5.1, 5.2, and 5.3, and updated the manuscript accordingly.
While Meta-Motivo shows strong imitation performance, **RGSD surpasses it in both skill diversity and downstream task performance**.
Reviewer 4 also noted that they would be willing to increase their score if this experiment was added during the rebuttal, which we have now completed.

---

**3. Newly added video results**

The original submission did not include a video. Reviewer 1 requested qualitative demonstrations of the learned skills.
We have now added a **4-minute video** illustrating:

* Learned skills of RGSD and baseline algorithms.
* Behaviors of high-level policies in downstream tasks (for both RGSD and CALM)
* Test-time control over motion diversity via latent-sampling variance
* RGSD applied to a multi-skill, unsegmented motion dataset
* Unconstrained skill generation by sampling latents far from motion embeddings

---

**4. Latent-space visualization confirming within-motion alignment**

Reviewers 1 and 2 requested empirical visualization of the latent space. Although the paper included a theoretical proof, reviewers questioned whether the alignment property could be achieved in practice given neural network limitations.
We now provide **empirical latent-space visualizations** showing that the alignment holds in practice, added in Appendix E.

---

**5. New experiments demonstrating test-time diversity modulation**

We realized that one major advantage of RGSD was not clearly highlighted in the original submission: after training, RGSD enables **test-time control over behavioral diversity** by adjusting the concentration parameter 𝜅 of the sampling distribution for the latent vector 𝑧. We now present **qualitative examples** illustrating how varying 𝜅 modulates behavioral diversity, along with **quantitative evidence** showing that the generated motions remain semantically consistent with the original reference motion.
These new results have been incorporated into the revised manuscript in **section 5.2** and **Appendix A**.

---

### Meta-Review · Area_Chair_fkeH · 2026-01-01

**Summary:**

This paper proposes Reference-Grounded Skill Discovery (RGSD), a framework for scaling unsupervised skill discovery to high-degree-of-freedom agents by grounding exploration in a semantically meaningful latent space learned from reference motion data. The method uses contrastive pretraining to embed each reference trajectory as a distinct direction on a unit hypersphere and then jointly performs imitation and discovery by sampling latents around these directions during reinforcement learning. Experiments on a 69-DoF SMPL humanoid demonstrate that RGSD can faithfully reproduce complex reference motions, discover diverse yet semantically related variations, and support downstream high-level control tasks that respect style constraints, outperforming prior imitation-based and unsupervised skill discovery baselines.

The main strengths of the work are its clear motivation, elegant geometric formulation of the latent space, and strong empirical results on a challenging high-DoF humanoid domain. The rebuttal substantially strengthened the paper by adding missing baselines (notably Meta-Motivo), expanding downstream evaluations to multiple tasks and styles, providing extensive qualitative video results, and empirically validating key theoretical assumptions such as within-motion alignment. The authors also appropriately clarified the scope of “discovery,” toning down overstated claims and explicitly positioning RGSD as discovering semantically related variations rather than fully compositional or primitive-level skills. Remaining weaknesses include limited generality beyond reference-covered behaviors, reliance on curated reference data, and downstream tasks that are still somewhat tailored to the latent structure, but these issues were largely acknowledged and framed as future work. Overall, the paper makes a solid and well-supported contribution to reference-guided skill discovery, so I recommend accepting this paper.

**Reviewer Concerns:**

Several major concerns raised by reviewers were convincingly addressed in the rebuttal. The authors added multiple downstream tasks with clear quantitative and qualitative comparisons, resolving the issue of insufficient evaluation. Requests for additional baselines were addressed by incorporating Meta-Motivo across all experiments, thereby strengthening the empirical foundation of the method. Concerns about the realism of theoretical assumptions, particularly within-motion alignment, were mitigated through new empirical analyses and visualizations showing near-perfect alignment in practice. The lack of qualitative demonstrations was fully resolved by the addition of extensive videos, and questions about semantic novelty were clarified by explicitly redefining the scope of discovery and providing controlled experiments that illustrate diversity modulation and limited novelty without overclaiming compositional generalization.

Some concerns remain partially outstanding but are less critical. The method still assumes reasonably well-curated reference motions and does not handle primitive disentanglement or true compositional generalization, which some reviewers initially expected. While the authors correctly clarified that these capabilities are out of scope, this does limit the broader applicability of the approach. Additionally, although the authors justified their evaluation choices (e.g., FID and downstream task design), questions about generalization to noisier, larger-scale, or less structured reference datasets remain open and are deferred to future work rather than fully resolved.

**Reviewer Scores:**

Based on the discussion, it is likely that at least one initially negative reviewer (e.g., Reviewer a8F6) would have increased their score modestly, from a clear reject to a borderline or weak reject/neutral, given that many empirical and clarity-related concerns were addressed, even if their expectations about compositionality were ultimately deemed out of scope. Reviewers who were already borderline (e.g., Reviewers 8xWQ and 1tD4) would likely have shifted to a weak accept, as the rebuttal directly addressed missing baselines, empirical validation, and overclaiming. The most positive reviewer (Yhxb) would likely maintain an accept score. Overall, the post-rebuttal trajectory suggests a shift from mixed to negative toward a consensus of cautious acceptance.

---

> ### Public Comment · ~Seungeun_Rho1 · 2026-02-27
>
> Dear Area Chair,
>
> Thank you very much for your thoughtful meta-review. We truly appreciate the time and care you took to synthesize the discussion.
> We are especially grateful that you recognized the core contribution of RGSD and the improvements made during the rebuttal process. Your feedback will also help guide our future work on broader generalization and scalability.
>
> Thank you again for your support.
>
> Sincerely,
> The Authors

---

### Decision · Program_Chairs · 2026-01-26

Accept (Poster)